# DIFFUSION MODELS ALREADY HAVE A SEMANTIC LATENT SPACE

**Mingi Kwon, Jaeseok Jeong, Youngjung Uh**[*]
Department of Artificial Intelligence
Yonsei University
Seoul, Republic of Korea
{kwonmingi,jete_jeong,yj.uh}@yonsei.ac.kr

## ABSTRACT

Diffusion models achieve outstanding generative performance in various domains. Despite their great success, they lack *semantic* latent space which is essential for controlling the generative process. To address the problem, we propose **asy**mmetric **r**everse **p**rocess (Asyrp) which discovers the semantic latent space in *frozen* pretrained diffusion models. Our semantic latent space, named *h-space*, has nice properties to accommodate semantic image manipulation: homogeneity, linearity, robustness, and consistency across timesteps. In addition, we introduce a principled design of the generative process for versatile editing and quality boosting by quantifiable measures: *editing strength* of an interval and *quality deficiency* at a timestep. Our method is applicable to various architectures (DDPM++, iDDPM, and ADM) and datasets (CelebA-HQ, AFHQ-dog, LSUN-church, LSUN-bedroom, and METFACES). Project page: https://kwonminki.github.io/Asyrp/

## 1 INTRODUCTION

In image synthesis, diffusion models have advanced to achieve state-of-the-art performance regarding quality and mode coverage since the introduction of denoising diffusion probabilistic models (Ho et al., 2020). They disrupt images by adding noise through multiple steps of forward process and generate samples by progressive denoising through multiple steps of reverse (i.e., generative) process. Since their deterministic version provides nearly perfect reconstruction of original images (Song et al., 2020a), they are suitable for image editing, which renders target attributes on the real images. However, simply editing the latent variables (i.e., intermediate noisy images) causes degraded results (Kim & Ye, 2021). Instead, they require complicated procedures: providing guidance in the reverse process or finetuning models for an attribute.

Figure 1(a-c) briefly illustrates the existing approaches. Image guidance mixes the latent variables of the guiding image with unconditional latent variables (Choi et al., 2021; Lugmayr et al., 2022; Meng et al., 2021). Though it provides some control, it is ambiguous to specify which attribute to reflect among the ones in the guide and the unconditional result, and it lacks intuitive control for the magnitude of change. Classifier guidance manipulates images by imposing gradients of a classifier on the latent variables in the reverse process to match the target class (Dhariwal & Nichol, 2021; Avrahami et al., 2022; Liu et al., 2021). It requires training an extra classifier for the latent variables, i.e., noisy images. Furthermore, computing gradients through the classifier during sampling is costly. Finetuning the whole model can steer the resulting images to the target attribute without the above problems (Kim & Ye, 2021). Still, it requires multiple models to reflect multiple descriptions.

On the other hand, generative adversarial networks (Goodfellow et al., 2020) inherently provide straightforward image editing in their latent space. Given a latent vector for an original image, we can find the direction in the latent space that maximizes the similarity of the resulting image with a target description in CLIP embedding (Patashnik et al., 2021). The latent direction found on one

---

[*]corresponding author

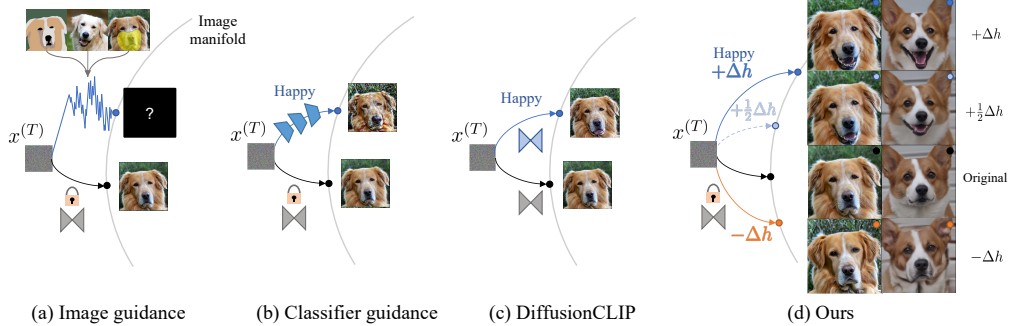

(a) Image guidance     (b) Classifier guidance     (c) DiffusionCLIP     (d) Ours

Figure 1: **Manipulation approaches for diffusion models.** (a) Image guidance suffers ambiguity while controlling the generative process. (b) Classifier guidance requires an extra classifier, is hardly editable, degrades quality, or alters the content. (c) DiffusionCLIP requires fine-tuning the whole model. (d) Our method discovers a semantic latent space of a *frozen* diffusion model.

image leads to the same manipulation of other images. However, given a *real* image, finding its exact latent vector is often challenging and produces unexpected appearance changes.

It would allow admirable image editing if the diffusion models with the nearly perfect inversion property have such a semantic latent space. Preechakul et al. (2022) introduces an additional input to the reverse diffusion process: a latent vector from an original image embedded by an extra encoder. This latent vector contains the semantics to condition the process. However, it requires training from scratch and does not match with pretrained diffusion models.

In this paper, we propose an **asy**mmetric **r**everse **p**rocess (*Asyrp*) which discovers the semantic latent space of a *frozen* diffusion model such that modifications in the space edits attributes of the original images. Our semantic latent space, named *h-space*, has the properties necessary for editing applications as follows. The same shift in this space results in the same attribute change in all images. Linear changes in this space lead to linear changes in attributes. The changes do not degrade the quality of the resulting images. The changes throughout the timesteps are almost identical to each other for a desired attribute change. Figure 1(d) illustrates some of these properties and § 5.3 provides detailed analyses. To the best of our knowledge, it is the first attempt to discover the semantic latent space in the frozen pretrained diffusion models. Spoiler alert: our semantic latent space is different from the intermediate latent variables in the diffusion process. Moreover, we introduce a principled design of the generative process for versatile editing and quality boosting by quantifiable measures: *editing strength* of an interval and *quality deficiency* at a timestep. Extensive experiments demonstrate that our method is generally applicable to various architectures (DDPM++, iDDPM, and ADM) and datasets (CelebA-HQ, AFHQ-dog, LSUN-church, LSUN-bedroom, and MᴇᴛFᴀᴄᴇs).

## 2 BACKGROUND

We briefly describe essential backgrounds. The rest of the related work is deferred to Appendix A.

### 2.1 DENOISING DIFFUSION PROBABILITY MODEL (DDPM)

DDPM is a latent variable model that learns a data distribution by denoising noisy images (Ho et al., 2020). The forward process diffuses the data samples through Gaussian transitions parameterized with a Markov process:

$$q\left(\boldsymbol{x}_t \mid \boldsymbol{x}_{t-1}\right) = \mathcal{N}\left(\boldsymbol{x}_t; \sqrt{1-\beta_t}\boldsymbol{x}_{t-1}, \beta_t \mathbf{I}\right) = \mathcal{N}\left(\sqrt{\frac{\alpha_t}{\alpha_{t-1}}}\boldsymbol{x}_{t-1}, \left(1 - \frac{\alpha_t}{\alpha_{t-1}}\right)\mathbf{I}\right), \quad (1)$$

where $\{\beta_t\}_{t=1}^{T}$ is the variance schedule and $\alpha_t = \prod_{s=1}^{t}(1-\beta_s)$. Then the reverse process becomes $p_\theta\left(\boldsymbol{x}_{0:T}\right) := p\left(\boldsymbol{x}_T\right) \prod_{t=1}^{T} p_\theta\left(\boldsymbol{x}_{t-1} \mid \boldsymbol{x}_t\right)$, starting from $\boldsymbol{x}_T \sim \mathcal{N}(0, \mathbf{I})$ with noise predictor $\boldsymbol{\epsilon}_t^\theta$:

$$\boldsymbol{x}_{t-1} = \frac{1}{\sqrt{1-\beta_t}}\left(\boldsymbol{x}_t - \frac{\beta_t}{\sqrt{1-\alpha_t}}\boldsymbol{\epsilon}_t^\theta\left(\boldsymbol{x}_t\right)\right) + \sigma_t \boldsymbol{z}_t, \quad (2)$$

where $\boldsymbol{z}_t \sim \mathcal{N}(0, \mathbf{I})$ and $\sigma_t^2$ is a variance of the reverse process which is set to $\sigma_t^2 = \beta_t$ by DDPM.

## 2.2 Denoising Diffusion Implicit Model (DDIM)

DDIM redefines Eq. (1) as $q_\sigma(\boldsymbol{x}_{t-1}|\boldsymbol{x}_t, \boldsymbol{x}_0) = \mathcal{N}(\sqrt{\alpha_{t-1}}\boldsymbol{x}_0 + \sqrt{1 - \alpha_{t-1} - \sigma_t^2} \cdot \frac{\boldsymbol{x}_t - \sqrt{\alpha_t}\boldsymbol{x}_0}{\sqrt{1-\alpha_t}}, \sigma_t^2 \boldsymbol{I})$ which is a non-Markovian process (Song et al., 2020a). Accordingly, the reverse process becomes

$$\boldsymbol{x}_{t-1} = \sqrt{\alpha_{t-1}} \underbrace{\left( \frac{\boldsymbol{x}_t - \sqrt{1 - \alpha_t}\boldsymbol{\epsilon}_t^\theta(\boldsymbol{x}_t)}{\sqrt{\alpha_t}} \right)}_{\text{"predicted } \boldsymbol{x}_0\text{ "}} + \underbrace{\sqrt{1 - \alpha_{t-1} - \sigma_t^2} \cdot \boldsymbol{\epsilon}_t^\theta(\boldsymbol{x}_t)}_{\text{"direction pointing to } \boldsymbol{x}_t\text{ "}} + \underbrace{\sigma_t \boldsymbol{z}_t}_{\text{random noise}}, \qquad (3)$$

where $\sigma_t = \eta\sqrt{(1 - \alpha_{t-1})/(1 - \alpha_t)}\sqrt{1 - \alpha_t/\alpha_{t-1}}$. When $\eta = 1$ for all $t$, it becomes DDPM. As $\eta = 0$, the process becomes deterministic and guarantees nearly perfect inversion.

## 2.3 Image manipulation with CLIP

CLIP learns multimodal embeddings with an image encoder $E_I$ and a text encoder $E_T$ whose similarity indicates semantic similarity between images and texts (Radford et al., 2021). Compared to directly minimizing the cosine distance between the edited image and the target description (Patashnik et al., 2021), directional loss with cosine distance achieves homogeneous editing without mode collapse (Gal et al., 2021):

$$\mathcal{L}_{\text{direction}}\left(\boldsymbol{x}^{\text{edit}}, y^{\text{target}}; \boldsymbol{x}^{\text{source}}, y^{\text{source}}\right) := 1 - \frac{\Delta I \cdot \Delta T}{\|\Delta I\|\|\Delta T\|}, \qquad (4)$$

where $\Delta T = E_T\left(y^{\text{target}}\right) - E_T\left(y^{\text{source}}\right)$ and $\Delta I = E_I\left(\boldsymbol{x}^{\text{edit}}\right) - E_I\left(\boldsymbol{x}^{\text{source}}\right)$ for edited image $\boldsymbol{x}^{\text{edit}}$, target description $y^{\text{target}}$, original image $\boldsymbol{x}^{\text{source}}$, and source description $y^{\text{source}}$. We use the prompts 'smiling face' and 'face' as the target and source descriptions for facial attribute `smiling`.

## 3 Discovering semantic latent space in diffusion models

This section explains why naive approaches do not work and proposes a new controllable reverse process. Then we describe the techniques for controlling the generative process. Throughout this paper, we use an abbreviated version of Eq. (3):

$$\boldsymbol{x}_{t-1} = \sqrt{\alpha_{t-1}}\,\mathbf{P}_t(\boldsymbol{\epsilon}_t^\theta(\boldsymbol{x}_t)) + \mathbf{D}_t(\boldsymbol{\epsilon}_t^\theta(\boldsymbol{x}_t)) + \sigma_t \boldsymbol{z}_t, \qquad (5)$$

where $\mathbf{P}_t(\boldsymbol{\epsilon}_t^\theta(\boldsymbol{x}_t))$ denotes the predicted $\boldsymbol{x}_0$ and $\mathbf{D}_t(\boldsymbol{\epsilon}_t^\theta(\boldsymbol{x}_t))$ denotes the direction pointing to $\boldsymbol{x}_t$. We omit $\sigma_t \boldsymbol{z}_t$ for brevity, except when $\eta \neq 0$. We further abbreviate $\mathbf{P}_t(\boldsymbol{\epsilon}_t^\theta(\boldsymbol{x}_t))$ as $\mathbf{P}_t$ and $\mathbf{D}_t(\boldsymbol{\epsilon}_t^\theta(\boldsymbol{x}_t))$ as $\mathbf{D}_t$ when the context clearly specifies the arguments.

### 3.1 Problem

We aim to allow semantic latent manipulation of images $\boldsymbol{x}_0$ generated from $\boldsymbol{x}_T$ given a *pretrained and frozen* diffusion model. The easiest idea to manipulate $\boldsymbol{x}_0$ is simply updating $\boldsymbol{x}_T$ to optimize the directional CLIP loss given text prompts with Eq. (4). However, it leads to distorted images or incorrect manipulation (Kim & Ye, 2021).

An alternative approach is to shift the noise $\boldsymbol{\epsilon}_t^\theta$ predicted by the network at each sampling step. However, it does not achieve manipulating $\boldsymbol{x}_0$ because the intermediate changes in $\mathbf{P}_t$ and $\mathbf{D}_t$ cancel out each other resulting in the same $p_\theta(\boldsymbol{x}_{0:T})$, similarly to destructive interference.

**Theorem 1.** *Let $\boldsymbol{\epsilon}_t^\theta$ be a predicted noise during the original reverse process at $t$ and $\tilde{\boldsymbol{\epsilon}}_t^\theta$ be its shifted counterpart. Then, $\Delta\boldsymbol{x}_t = \tilde{\boldsymbol{x}}_{t-1} - \boldsymbol{x}_{t-1}$ is negligible where $\tilde{\boldsymbol{x}}_{t-1} = \sqrt{\alpha_{t-1}}\,\mathbf{P}_t(\tilde{\boldsymbol{\epsilon}}_t^\theta(\boldsymbol{x}_t)) + \mathbf{D}_t(\tilde{\boldsymbol{\epsilon}}_t^\theta(\boldsymbol{x}_t))$. I.e., the shifted terms of $\tilde{\boldsymbol{\epsilon}}_t^\theta$ in $\mathbf{P}_t$ and $\mathbf{D}_t$ destruct each other in the reverse process.*

Appendix C proves above theorem. Figure 13(a-b) shows that $\tilde{\boldsymbol{x}}_0$ is almost identical to $\boldsymbol{x}_0$.

### 3.2 Asymmetric reverse process

In order to break the interference, we propose a new controllable reverse process with asymmetry:

$$\boldsymbol{x}_{t-1} = \sqrt{\alpha_{t-1}}\,\mathbf{P}_t(\tilde{\boldsymbol{\epsilon}}_t^\theta(\boldsymbol{x}_t)) + \mathbf{D}_t(\boldsymbol{\epsilon}_t^\theta(\boldsymbol{x}_t)), \qquad (6)$$

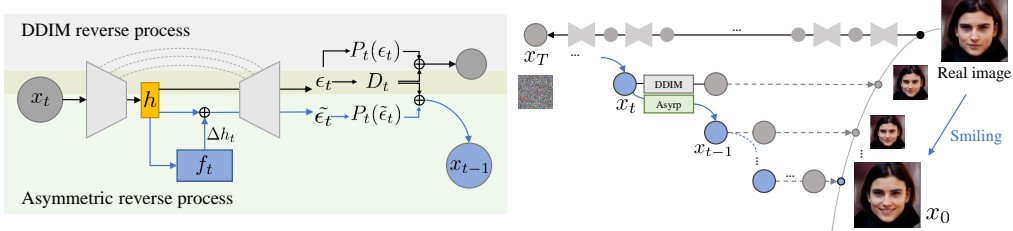

Figure 2: **Generative process of Asyrp.** The green box on the left illustrates Asyrp which only alters $\mathbf{P}_t$ while preserving $\mathbf{D}_t$ shared by DDIM. The right describes that Asyrp modifies the original reverse process toward the target attribute reflecting the change in *h-space*.

i.e., we modify only $\mathbf{P}_t$ by shifting $\boldsymbol{\epsilon}_t^\theta$ to $\tilde{\boldsymbol{\epsilon}}_t^\theta$ while preserving $\mathbf{D}_t$. Intuitively, it modifies the original reverse process according to $\Delta\boldsymbol{\epsilon}_t = \tilde{\boldsymbol{\epsilon}}_t^\theta - \boldsymbol{\epsilon}_t^\theta$ while it does not alter the direction toward $\boldsymbol{x}_t$ so that $\boldsymbol{x}_{t-1}$ follows the original flow $\mathbf{D}_t$ at each sampling step. Figure 2 illustrates the above intuition.

As in Avrahami et al. (2022), we use the modified $\mathbf{P}_t^{\text{edit}}$ and the original $\mathbf{P}_t^{\text{source}}$ as visual inputs for the directional CLIP loss in Eq. (4), and regularize the difference between the modified $\mathbf{P}_t^{\text{edit}}$ and the original $\mathbf{P}_t^{\text{source}}$. We find $\Delta\boldsymbol{\epsilon} = \arg\min_{\Delta\boldsymbol{\epsilon}} \mathbb{E}_t \mathcal{L}^{(t)}$ where

$$\mathcal{L}^{(t)} = \lambda_{\text{CLIP}} \mathcal{L}_{\text{direction}} \left( \mathbf{P}_t^{\text{edit}}, y^{\text{ref}}; \mathbf{P}_t^{\text{source}}, y^{\text{source}} \right) + \lambda_{\text{recon}} \left| \mathbf{P}_t^{\text{edit}} - \mathbf{P}_t^{\text{source}} \right| \tag{7}$$

Although $\Delta\boldsymbol{\epsilon}$ indeed renders the attribute in the $\boldsymbol{x}_0^{\text{edit}}$, $\boldsymbol{\epsilon}$-space lacks the necessary properties of the semantic latent space in diffusion models that will be described in the following.

### 3.3 *h-space*

Note that $\boldsymbol{\epsilon}_t^\theta$ is implemented as U-Net in all state-of-the-art diffusion models. We choose its bottleneck, the deepest feature maps $\boldsymbol{h}_t$, to control $\boldsymbol{\epsilon}_t^\theta$. By design, $\boldsymbol{h}_t$ has smaller spatial resolutions and high-level semantics than $\boldsymbol{\epsilon}_t^\theta$. Accordingly, the sampling equation becomes

$$\boldsymbol{x}_{t-1} = \sqrt{\alpha_{t-1}} \, \mathbf{P}_t(\boldsymbol{\epsilon}_t^\theta(\boldsymbol{x}_t | \Delta\boldsymbol{h}_t)) + \mathbf{D}_t(\boldsymbol{\epsilon}_t^\theta(\boldsymbol{x}_t)) + \sigma_t \boldsymbol{z}_t, \tag{8}$$

where $\boldsymbol{\epsilon}_t^\theta(\boldsymbol{x}_t | \Delta\boldsymbol{h}_t)$ adds $\Delta\boldsymbol{h}_t$ to the original feature maps $\boldsymbol{h}_t$. The $\Delta\boldsymbol{h}_t$ minimizing the same loss in Eq. (7) with $\mathbf{P}_t(\boldsymbol{\epsilon}_t^\theta(\boldsymbol{x}_t | \Delta\boldsymbol{h}_t))$ instead of $\mathbf{P}_t(\tilde{\boldsymbol{\epsilon}}_t^\theta(\boldsymbol{x}_t))$ successfully manipulates the attributes.

We observe that *h-space* in Asyrp has the following properties that others do not have.

- The same $\Delta\boldsymbol{h}$ leads to the same effect on different samples.
- Linearly scaling $\Delta\boldsymbol{h}$ controls the magnitude of attribute change, even with negative scales.
- Adding multiple $\Delta\boldsymbol{h}$ manipulates the corresponding multiple attributes simultaneously.
- $\Delta\boldsymbol{h}$ preserves the quality of the resulting images without degradation.
- $\Delta\boldsymbol{h}_t$ is roughly consistent across different timesteps $t$.

The above properties are demonstrated thoroughly in § 5.3. Appendix D.3 provides details of *h-space* and suboptimal results from alternative choices.

### 3.4 Implicit neural directions

Although $\Delta\boldsymbol{h}$ succeeds in manipulating images, directly optimizing $\Delta\boldsymbol{h}_t$ on multiple timesteps requires many iterations of training with a carefully chosen learning rate and its scheduling. Instead, we define an implicit function $\boldsymbol{f}_t(\boldsymbol{h}_t)$ which produces $\Delta\boldsymbol{h}_t$ for given $\boldsymbol{h}_t$ and $t$. $\boldsymbol{f}_t$ is implemented as a small neural network with two $1 \times 1$ convolutions concatenating timestep $t$. See Appendix E for the details. Accordingly, we optimize the same loss in Eq. (7) with $\mathbf{P}_t^{\text{edit}} = \mathbf{P}_t(\boldsymbol{\epsilon}_t^\theta(\boldsymbol{x}_t | \boldsymbol{f}_t))$.

Learning $\boldsymbol{f}_t$ is more robust to learning rate settings and converges faster than learning every $\Delta\boldsymbol{h}_t$. In addition, as $\boldsymbol{f}_t$ learns an implicit function for given timesteps and bottleneck features, it generalizes

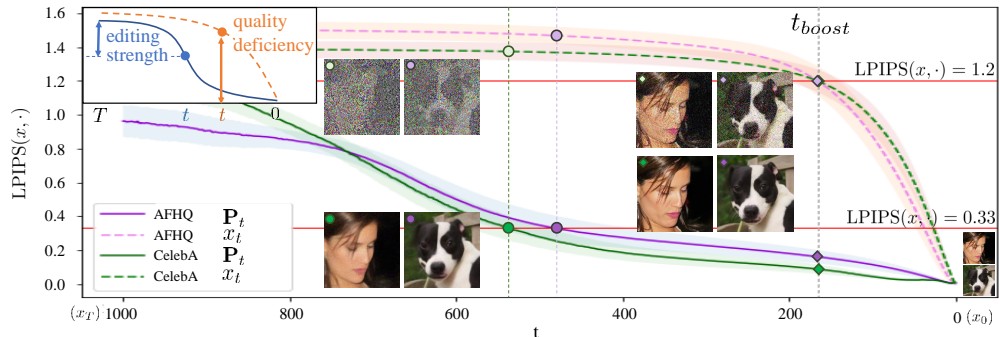

Figure 3: **Intuition for choosing the intervals for editing and quality boosting.** We choose the intervals by quantifying two measures (top left inset). The editing strength of an interval $[T, t]$ measures its perceptual difference from $T$ until $t$. We set $[T, t]$ to the interval with the smallest editing strength that synthesizes $\mathbf{P}_t$ close to $x$, i.e., $\text{LPIPS}(x, \mathbf{P}_t) = 0.33$. Editing flexibility of an interval $[t, 0]$ measures the potential amount of changes after $t$. Quality deficiency at $t$ measures the amount of noise in $x_t$. We set $[t, 0]$ to handle large quality deficiency (i.e., $\text{LPIPS}(x, x_t) = 1.2$) with small editing flexibility.

to unseen timesteps and bottleneck features. The generalization allows us to borrow the accelerated training scheme of DDIM defined on a subsequence $\{x_{\tau_i}\}_{\forall i \in [1, S]}$ where $\{\tau_i\}$ is a subsequence of $[1, ..., T]$ and $S < T$. Then, we can use the generative process with a custom subsequence $\{\tilde{\tau}_i\}$ with length $\tilde{S} < T$ through normalization: $\Delta \tilde{\boldsymbol{h}}_{\tilde{\tau}} = \boldsymbol{f}_{\tilde{\tau}}(\boldsymbol{h}_{\tilde{\tau}}) S / \tilde{S}$. It preserves the amount of $\sum \Delta \boldsymbol{h}_t$, $\Delta \tilde{\boldsymbol{h}}_{\tilde{\tau}} \tilde{S} = \Delta \boldsymbol{h}_t S$. Therefore, we can use $\boldsymbol{f}_t$ trained on any subsequence for any length of the generative process. See Appendix F for details. We use $\boldsymbol{f}_t$ to get $\Delta \boldsymbol{h}_t$ for all experiments except Figure 6.

## 4 GENERATIVE PROCESS DESIGN

This section describes the entire editing process, which consists of three phases: editing with Asyrp, traditional denoising, and quality boosting. We design formulas to determine the length of each phase with quantifiable measures.

### 4.1 EDITING PROCESS WITH ASYRP

Diffusion models generate the high-level context in the early stage and imperceptible fine details in the later stage (Choi et al., 2022). Likewise, we modify the generative process in the early stage to achieve semantic changes. We refer to the early stage as the editing interval $[T, t_{\text{edit}}]$.

$\text{LPIPS}(x, \mathbf{P}_T)$ and $\text{LPIPS}(x, \mathbf{P}_t)$ calculate the perceptual distance between the original image and the predicted image at time steps $T$ and $t$, respectively. Intuitively, the high-level content is already determined by the predicted terms at the respective timesteps and LPIPS measures the remaining component to be edited through the remaining reverse process. Consequently, we define *editing strength* of an interval $[T, t]$:

$$\xi_t = \text{LPIPS}(x, \mathbf{P}_T) - \text{LPIPS}(x, \mathbf{P}_t)$$

indicating the perceptual change from timestep $T$ to $t$ in the original generative process. Figure 3 illustrates $\text{LPIPS}(x, \cdot)$ for $\mathbf{P}_t$ and $x_t$ with examples and the inset depicts *editing strength*. The shorter editing interval has the lower $\xi_t$, and the longer editing interval brings more changes to the resulting images. We seek the shortest editing interval which will bring enough distinguishable changes in the images in general. We empirically find that $t_{\text{edit}}$ with $\text{LPIPS}(x, \mathbf{P}_{t_{\text{edit}}}) = 0.33$ builds the shortest editing interval with enough editing strength as $\mathbf{P}_{t_{\text{edit}}}$ has nearly all visual attributes in $x$.

However, some attributes require more visual changes than others, e.g., `pixar > smile`. For such attributes, we increase the editing strength $\xi_t$ by $\delta = 0.33d(E_T(y_{\text{source}}), E_T(y_{\text{target}}))$ where

$E_T(\cdot)$ produces CLIP text embedding, $y_{(\cdot)}$ denotes the descriptions, and $d(\cdot, \cdot)$ computes the cosine distance between the arguments. Choosing $t_{\text{edit}}$ with $\text{LPIPS}(\boldsymbol{x}, \mathbf{P}_{t_{\text{edit}}}) = 0.33 - \delta$ expands the editing interval to a suitable length. It consistently produces good results in various settings. The supporting experiments are shown in Appendix G.

## 4.2 QUALITY BOOSTING WITH STOCHASTIC NOISE INJECTION

Although DDIM achieves nearly perfect inversion by removing stochasticity ($\eta = 0$), Karras et al. (2022) demonstrate that stochasticity improves image quality. Likewise, we inject stochastic noise in the boosting interval $[t_{\text{boost}}, 0]$.

Though the longer boosting interval would achieve higher quality, boosting over excessively long intervals would modify the content. Hence, we want to determine the shortest interval that shows enough quality boosting to guarantee minimal change in the content. We consider the noise in the image as the capacity for the quality boosting and define *quality deficiency* at $t$: $\gamma_t = \text{LPIPS}(\boldsymbol{x}, \boldsymbol{x}_t)$ indicating the amount of noise in $\boldsymbol{x}_t$ compared to the original image. We use $\boldsymbol{x}_t$ instead of $\mathbf{P}_t$ because we consider the actual image rather than the semantics. Figure 3 inset depicts editing flexibility and quality deficiency. We empirically find that $t_{\text{boost}}$ with $\gamma_{t_{\text{boost}}} = 1.2$ achieves quality boosting with minimal content change. We confirmed that the editing strength of the intervals $[t_{\text{boost}}, 0]$ is guaranteed to be less than 0.25. In Figure 3, after $t_{\text{boost}}$, $\text{LPIPS}(\boldsymbol{x}, \boldsymbol{x}_t)$ sharply drops in the original generative process while $\text{LPIPS}(\boldsymbol{x}, \mathbf{P}_t)$ changes little. Note that most of the quality degradation of the resulting images is caused by DDIM reverse process, not by Asyrp. We use this quality boosting for all experiments except ablation in Appendix H.

## 4.3 OVERALL PROCESS OF IMAGE EDITING

Using $t_{\text{edit}}$ and $t_{\text{boost}}$ determined by the above formulas, we modify the generative process of DDIM with

$$p_\theta^{(t)}(\boldsymbol{x}_{t-1} \mid \boldsymbol{x}_t) = \begin{cases} \mathcal{N}\left(\sqrt{\alpha_{t-1}}\,\mathbf{P}_t(\boldsymbol{\epsilon}_t^\theta(\boldsymbol{x}_t|\boldsymbol{f}_t)) + \mathbf{D}_t, \sigma_t^2 \boldsymbol{I}\right), \; \eta = 0 & \text{if } T \geq t \geq t_{\text{edit}} \\ \mathcal{N}\left(\sqrt{\alpha_{t-1}}\,\mathbf{P}_t(\boldsymbol{\epsilon}_t^\theta(\boldsymbol{x}_t)) + \mathbf{D}_t, \sigma_t^2 \boldsymbol{I}\right), \; \eta = 0 & \text{if } t_{\text{edit}} > t \geq t_{\text{boost}} \\ \mathcal{N}\left(\sqrt{\alpha_{t-1}}\,\mathbf{P}_t(\boldsymbol{\epsilon}_t^\theta(\boldsymbol{x}_t)) + \mathbf{D}_t, \sigma_t^2 \boldsymbol{I}\right), \; \eta = 1 & \text{if } t_{\text{boost}} > t \end{cases} \quad (9)$$

The visual overview and comprehensive algorithms of the entire process are in Appendix I.

## 5 EXPERIMENTS

In this section, we show the effectiveness of semantic latent editing in *h-space* with Asyrp on various attributes, datasets and architectures in § 5.1. Moreover, we provide quantitative results including user study in § 5.2. Lastly, we provide detailed analyses for the properties of the semantic latent space on *h-space* and alternatives in § 5.3.

**Implementation details.** We implement our method on various settings: CelebA-HQ (Karras et al., 2018) and LSUN-bedroom/-church (Yu et al., 2015) on DDPM++ (Song et al., 2020b) (Meng et al., 2021); AFHQ-dog (Choi et al., 2020) on iDDPM (Nichol & Dhariwal, 2021); and MET-FACES (Karras et al., 2020) on ADM with P2-weighting (Dhariwal & Nichol, 2021) (Choi et al., 2022). Please note that all models are official pretrained checkpoints and are kept frozen. Detailed settings including the coefficients for $\lambda_{\text{CLIP}}$ and $\lambda_{\text{recon}}$, and source/target descriptions can be found in Appendix J.1. We train $\boldsymbol{f}_t$ with $S = 40$ for 1 epoch using 1000 samples. The real samples are randomly chosen from each dataset for in-domain-like attributes. For out-of-domain-like attributes, we randomly draw 1,000 latent variables $\boldsymbol{x}_T \sim \mathcal{N}(0, \boldsymbol{I})$. Details are described in Appendix J.2. Training takes about 20 minutes with three RTX 3090 GPUs. All the images in the figures are not used for training. We set $\tilde{S} = 1,000$ for inference. The code is available at https://github.com/kwonminki/Asyrp_official

## 5.1 VERSATILITY OF *h-space* WITH ASYRP

Figure 4 shows the effectiveness of our method on various datasets and any existing U-Net based architectures. Our method can synthesize the attributes that are not even included in the training

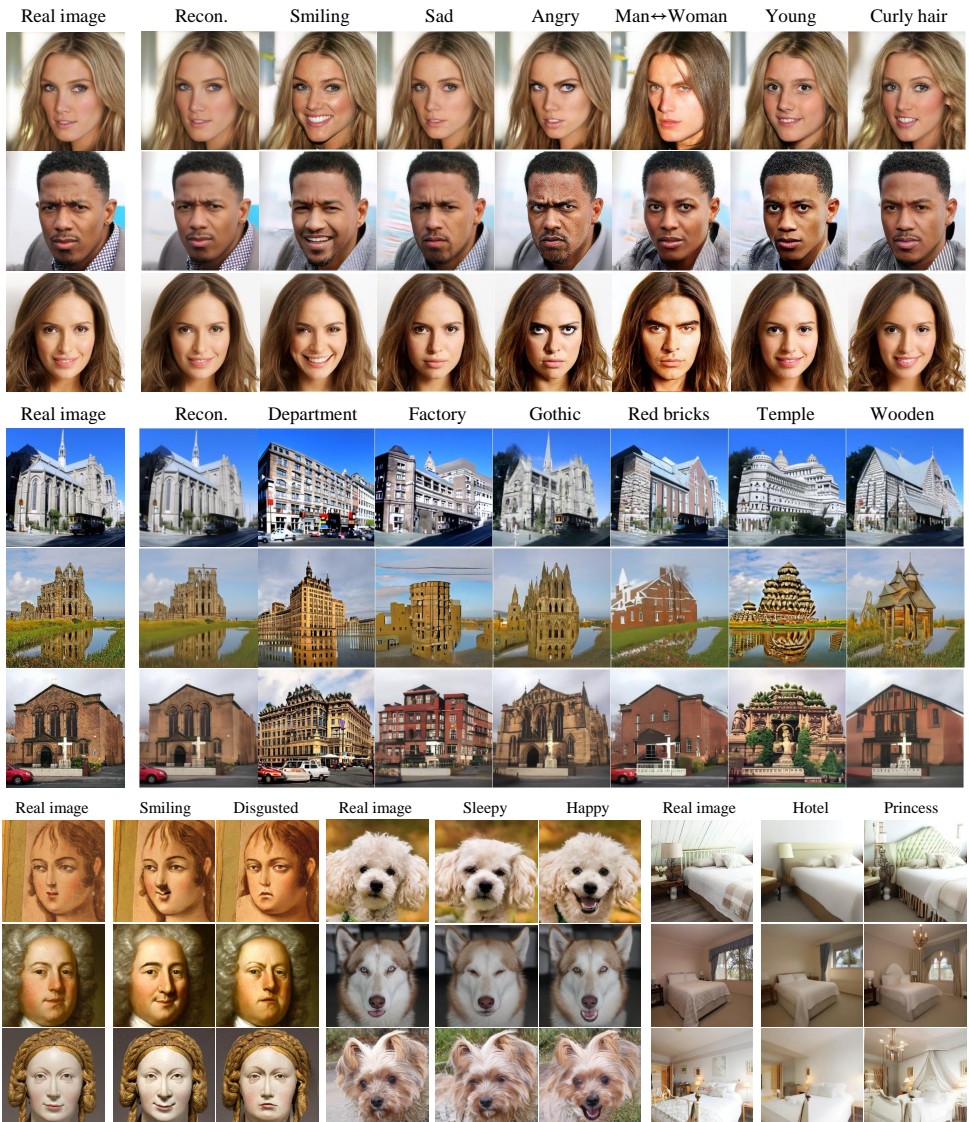

Figure 4: **Editing results of Asyrp on various datasets.** We conduct experiments on CelebA-HQ, LSUN-church, METFACES, AFHQ-dog, and LSUN-bedroom.

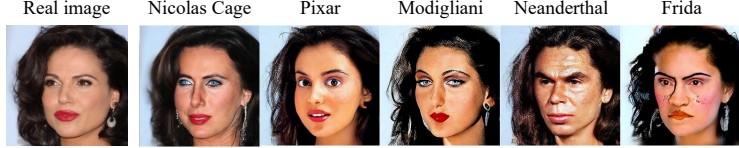

Figure 5: **Editing results of Asyrp for unseen domains in CelebA-HQ dataset.**

dataset, such as church → {department, factory, and temple}. Even for dogs, our method synthesizes smiling Poodle and Yorkshire, the species that barely smile in the dataset. Figure 5 provides results for changing human faces to different identities, painting styles, and ancient primates. More result can be found in Appendix N. Versatility of our method is surprising because we do *not* alter the models but only shift the bottleneck feature maps in *h-space* with Asyrp during inference.

| | CelebA-HQ in-domain | | | CelebA-HQ unseen-domain | | | LSUN-church | | | |
| --- | --- | --- | --- | --- | --- | --- | --- | --- | --- | --- |
| | quality | attribute | overall | quality | attribute | overall | quality | attribute | diversity | overall |
| Asyrp (ours) | **98.36%** | **88.13%** | **94.92%** | **71.56%** | **59.84%** | **63.13%** | **73.19%** | **71.81%** | **87.50%** | **76.81%** |
| DiffusionCLIP | 1.64% | 11.88% | 5.08% | 28.44% | 40.16% | 36.88% | 26.81% | 28.19% | 12.50% | 23.19% |

Table 1: **User study** with 80 participants. The details are described in § K.1

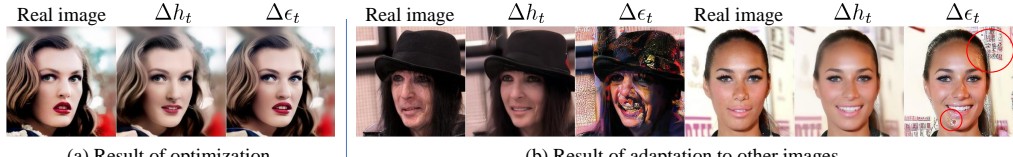

(a) Result of optimization · (b) Result of adaptation to other images

Figure 6: **Optimization for smiling on *h-space* and $\epsilon$-space.** (a) Optimizing $\Delta \boldsymbol{h}_t$ for a sample with smiling results in natural editing while the change due to optimizing $\Delta \boldsymbol{\epsilon}_t$ is relatively small. (b) Applying $\Delta \boldsymbol{h}_t$ from (a) to other samples yields the same attribute change while $\Delta \boldsymbol{\epsilon}_t$ distorts the images. The result of $\Delta \boldsymbol{\epsilon}_t$ is the best sample we could find.

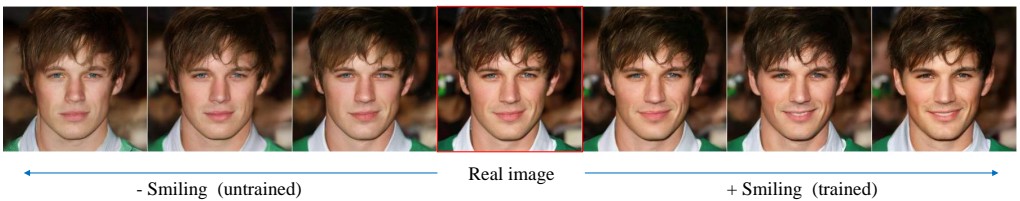

- Smiling (untrained) · Real image · + Smiling (trained)

Figure 7: **Linearity of *h-space*.** Linear interpolation and extrapolation on *h-space* lead to gradual changes even to the unseen intensity and directions. Right side shows the interpolation results by positive scaling of $\Delta h_t^{\text{smiling}}$. Left side shows the extrapolation results by negative scaling of $\Delta h_t^{smiling}$. Note that we did not train $\boldsymbol{f}_t$ for the negative direction.

## 5.2 QUANTITATIVE COMPARISON

Considering that our method can be combined with various diffusion models *without* finetuning, we do not find such a versatile competitor. Nonetheless, we compare Asyrp against DiffusionCLIP using the official code that edits the real images by finetuning the whole model. We asked 80 participants to choose the images with better quality, natural attribute change, and overall preference for given total of 40 sets of original images, ours, and DiffusionCLIP. Table 1 shows that Asyrp outperforms DiffusionCLIP in the all perspectives including the attributes unseen in the training dataset. We list the settings for fair comparison including the questions and example images in Appendix K.1. See § K.2 for more evaluation metrics: segmentation consistency (SC) and directional CLIP similarity($S_{\text{dir}}$).

## 5.3 ANALYSIS ON *h-space*

We provide detailed analyses to validate the properties of semantic latent space for diffusion models: homogeneity, linearity, robustness, and consistency across timesteps.

**Homogeneity.** Figure 6 illustrates homogeneity of *h-space* compared to $\epsilon$-space. One $\Delta \boldsymbol{h}_t$ optimized for an image results in the same attribute change to other input images. On the other hand, one $\Delta \boldsymbol{\epsilon}_t$ optimized for an image distorts other input images. In Figure 10, applying $\Delta \boldsymbol{h}_t^{\text{mean}} = \frac{1}{N} \sum \Delta \boldsymbol{h}_t^i$ produces almost identical results where $i$ indicates indices of $N = 20$ random samples.

**Linearity.** In Figure 7, we observe that linearly scaling a $\Delta \boldsymbol{h}$ reflects the amount of change in the visual attributes. Surprisingly, it generalizes to negative scales that are not seen during training. Moreover, Figure 8 shows that combinations of different $\Delta \boldsymbol{h}$'s yield their combined semantic changes in the resulting images. Appendix N.2 provides mixed interpolation between multiple attributes.

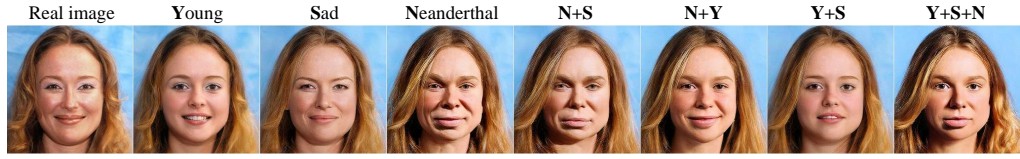

Figure 8: **Linear combination.** Combining multiple $\Delta h$s leads to combined attribute changes.

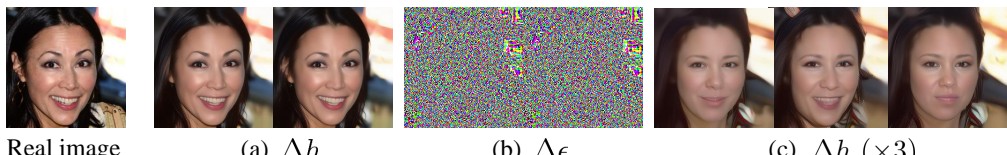

Figure 9: **Ramdom manipulation on *h-space* and $\epsilon$-space.** (a) Adding random noise with magnitude of $\Delta h_t^{\texttt{smiling}}$ and random direction in *h-space* leads to realistic images with small changes. (b) Adding random noise with magnitude of $\Delta \epsilon_t^{\texttt{smiling}}$ and random direction in $\epsilon$-space leads to severely distorted images. (c) Adding random noises with tripled magnitude produce diverse visual changes in realistic images.

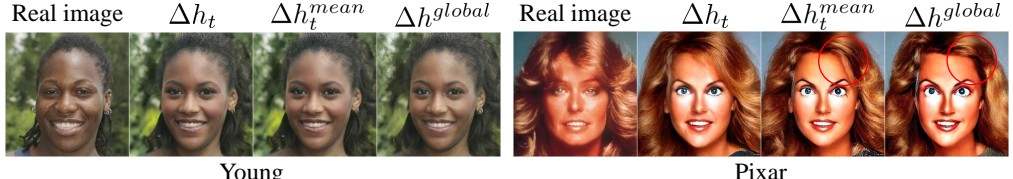

Figure 10: **Consistency on *h-space*.** Results of $\Delta h_t$, $\Delta h_t^{\text{mean}}$ and $\Delta h^{\text{global}}$ are almost identical for in-domain samples. However, we choose $\Delta h_t$ over others to prevent unexpected small difference in the unseen domain. We provide more results in § L.1.

**Robustness.** Figure 9 compares the effect of adding random noise in *h-space* and $\epsilon$-space. The random noises are chosen to be the vectors with random directions and magnitude of the example $\Delta h_t$ and $\Delta \epsilon_t$ in Figure 6 on each space. Perturbation in *h-space* leads to realistic images with a minimal difference or some semantic changes. On the contrary, perturbation in $\epsilon$-space severely distorts the resulting images. See Appendix D.2 for more analyses.

**Consistency across timesteps.** Recall that $\Delta h_t$ for all samples are homogeneous and replacing them by their mean $\Delta h_t^{\text{mean}}$ yields similar results. Interestingly, in Figure 10, we observe that adding a time-invariant $\Delta h^{\text{global}} = \frac{1}{T_e} \sum_t \Delta h_t^{\text{mean}}$ instead of $\Delta h_t$ also yields similar results where $T_e$ denotes the length of the editing interval $[T, t_{\text{edit}}]$. Though we use $\Delta h_t$ to deliver the best quality and manipulation, using $\Delta h_t^{\text{mean}}$ or even $\Delta h^{\text{global}}$ with some compromise would be worth trying for simplicity. We report more detail about mean and global direction in Appendix L.1.

## 6 CONCLUSION

We proposed a new generative process, Asyrp, which facilitates image editing in a semantic latent space *h-space* for pretrained diffusion models. *h-space* has nice properties as in the latent space of GANs: homogeneity, linearity, robustness, and consistency across timesteps. The full editing process is designed to achieve versatile editing and high quality by measuring editing strength and quality deficiency at timesteps. We hope that our approach and detailed analyses help cultivate a new paradigm of image editing in the semantic latent space of diffusion models. Combining previous finetuning or guidance techniques would be an interesting research direction.

ACKNOWLEDGMENTS

This work was supported by the National Research Foundation of Korea (NRF) grant funded by the Korea government (Ministry of Science and ICT) (No. 2021-0-00155)

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

# Appendix

## Table of Contents

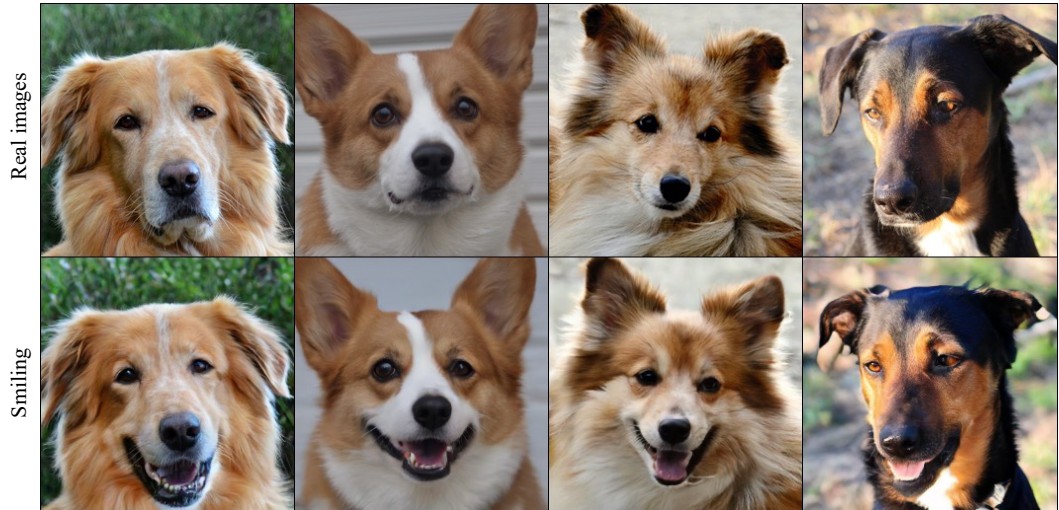

Figure 11: **Editing dog to smile.** We provide high-resolution results of editing real images in the teaser and more.

## A RELATED WORK

After Sohl-Dickstein et al. (2015), denoising diffusion probabilistic models (DDPMs) provide a universal approach for generative modeling (Ho et al., 2020). On the other hand, Song et al. (2020b) suggests score-based model and unifies SDEs incorporating diffusion models with score-based models. Subsequent works renovate diffusion models by focusing on architectures, scheduling, weighting, and fast sampling (Nichol & Dhariwal (2021), Karras et al. (2022), Choi et al. (2022), Song et al. (2020a), Watson et al. (2022)). They mainly consider random generation rather than controlled generation.

In the meantime, Dhariwal & Nichol (2021) introduces classifier guidance not only improving the quality of images but also retrieving specific class of images. Since it can apply any guidance, its variants have emerged (Sehwag et al. (2022), Avrahami et al. (2022), Liu et al. (2021), Nichol et al. (2021)). However it requires a noise-dependent classifier (or any off-the-shelf models) and additional cost to compute gradients for the guidance during its sampling process. The other works try to control the generative process using image-space guidance (Choi et al. (2021), Meng et al. (2021), Lugmayr et al. (2022), Avrahami et al. (2022)). They manipulate resulting images by matching noisy images with target images during the reverse process. Still, it is hard to expect delicate control of the reverse process from the image guidance. Furthermore, Preechakul et al. (2022) introduces an extra encoder which encodes the semantic features of a real image in order to condition the generative process. Although the semantics allow one to control diffusion models, it requires additional training from scratch with the encoder and inherently can not use the other pretrained diffusion models.

For controllability, Rombach et al. (2022) and Vahdat et al. (2021) apply another approach which adapts VAE (Kingma & Welling, 2013) and autoencoder (Rumelhart et al., 1985) to diffusion models. In spite of their great success in editing, their diffusion models learn the distribution of the learned embeddings in VAE or autoencoder, not the images. Kim & Ye (2021) proposes another strategy: fine-tuning a whole diffusion model for image editing. It shows valid performance but it requires each fine-tuned model corresponding each attribute.

In comparison, Asyrp enables outstanding manipulation without high computation, specifically designed architectures, or fine-tuning whole models.

Meanwhile, generative adversarial network Goodfellow et al. (2020) address their latent space for image editing (Ling et al. (2021), Härkönen et al. (2020), Chefer et al. (2021), Shen et al. (2020),

Yüksel et al. (2021), Patashnik et al. (2021), Gal et al. (2021), Dai et al. (2019), Xu et al. (2022)). However they have to conduct 'inversion' to their latent space for real image editing and 'GAN inversion' is often challenging and produces unexpected appearance changes.

On the contrary, Asyrp enables to use latent space of *real images* by nearly perfect easy inversion of DDIM.

## B    MORE DISCUSSION

In this section, we discuss the pros and cons of diffusion-model-based and GAN-based methods. And we provide guidelines for further improvements.

GAN-based latent manipulation methods Patashnik et al. (2021); Gal et al. (2021)require careful inversion from real images to latent codes for real image editing. On the contrary, our proposed method based on diffusion models has a powerful advantage; the sophisticated inversion method is not necessary. This means that we can obtain the latent code of an arbitrary real image even if the image is not in the trained domain. On the other hand, several inversion methods have been proposed for GANs to obtain the latent of the real image, and the corresponding latent manipulation method should be considered for each inversion method. For example, it is difficult to apply the method of editing in $w$ space to the method of inversion using $w^+$ space.

However, GANs have the advantage of fast sampling. In addition, diffusion models have a relatively slow sampling time. Additionally, we have to be aware of the time steps of diffusion models, which is still less well known.

The advantage of being free from Inversion provides the following milestones. The manipulation in the latent of the diffusion models is the same as the editing in real images. It can be expanded to segmentation, clustering, classification, etc. in *h-space* for real-world images.

It would be an interesting research direction to employ previous techniques. Our method can be used in conjunction with gradient guidance methods. Although we do not focus on random sampling, ours works effectively for sampling with stochastic. (See § M.) It may bring more diverse methods to steer diffusion models.

*h-space* in the latent diffusion models such as stable diffusion, is another interesting research direction. The main contribution of our paper is only modifying $\mathbf{P}_t$ while preserving $\mathbf{D}_t$, and can be adapted with latent diffusion models. However, since the latent meaning may be different due to structural differences, research on this is needed.

Furthermore, all of the properties of *h-space* according to the time step has not been fully discussed so far. Research on them can be expected to expand further.

**Limitations.**    Editing with Asyrp seldom yields changes in overall style or peripheral objects but edits attributes of the main object. Style transfer using frozen diffusion models is our future work.

**Societal impact / Ethics statement.**    Techniques for high-quality image manipulation such as Asyrp should be accompanied by social and/or technical solutions to prevent abuse. We acknowledge the potential ethical implications that may arise from the use of our image manipulation technique, Asyrp. We advocate for the development and implementation of social and technical solutions to prevent potential abuses such as spreading disinformation or propaganda. We are committed to ensuring fairness and non-discrimination, legal compliance, and research integrity in our work.

## C    PROOF OF THEOREM 1

*Proof* of **Theorem 1**. Let $\epsilon_t^\theta$ be a predicted noise during the original reverse process at $t$ and $\tilde{\epsilon}_t^\theta$ be its shifted counterpart. Then, $\Delta \boldsymbol{x}_t = \tilde{\boldsymbol{x}}_{t-1} - \boldsymbol{x}_{t-1}$ is negligible where $\tilde{\boldsymbol{x}}_{t-1} = \sqrt{\alpha_{t-1}}\, \mathbf{P}_t(\tilde{\epsilon}_t^\theta(\boldsymbol{x}_t)) + \mathbf{D}_t(\tilde{\epsilon}_t^\theta(\boldsymbol{x}_t))$.

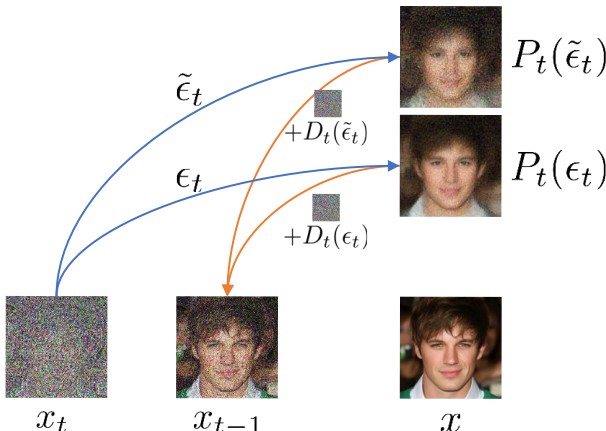

Figure 12: **Illustration of Theorem 1.** Upper blue line describes applying noise $\tilde{\epsilon}_t = \epsilon_t + \Delta\epsilon_t$ to produce $\mathbf{P}_t(\tilde{\epsilon}_t) =$ the shifted predicted $\boldsymbol{x}_0$. However, the shift due to $\Delta\epsilon_t$ is canceled out by the shift in $\mathbf{D}_t(\tilde{\epsilon}_t)$ due to $\Delta\epsilon_t$. As a results, applying $\Delta\epsilon_t$ both on $\mathbf{P}_t$ and $\mathbf{D}_t$ brings identical outputs to the original.

Define $\tilde{\boldsymbol{\epsilon}}_t^\theta(\boldsymbol{x}_t) = \boldsymbol{\epsilon}_t^\theta(\boldsymbol{x}_t) + \Delta\boldsymbol{\epsilon}_t$, $\{\beta_t\}_{t=1}^T = \{\beta_1 = \beta_{\min}, ..., \beta_T = \beta_{\max}\}$, and $\alpha_t = \prod_{s=1}^t (1 - \beta_s)$. Note that $\beta_{\max}$ is defined as a small value (e.g., $\beta_{\max} = 0.001$) and $\{\beta_t\}_{t=1}^T$ are defined by a decreasing schedule from $\beta_T = \beta_{\max}$ to $\beta_1 = \beta_{\min} \approx 0$ (e.g., $\beta_{\min} = 0.00001$). Then,

$$\tilde{\boldsymbol{x}}_{t-1} = \sqrt{\alpha_{t-1}}\, \mathbf{P}_t(\tilde{\boldsymbol{\epsilon}}_t^\theta(\boldsymbol{x}_t)) + \mathbf{D}_t(\tilde{\boldsymbol{\epsilon}}_t^\theta(\boldsymbol{x}_t)) \tag{10}$$

$$= \sqrt{\alpha_{t-1}} \left( \frac{\boldsymbol{x}_t - \sqrt{1-\alpha_t}\left(\boldsymbol{\epsilon}_t^\theta(\boldsymbol{x}_t) + \Delta\boldsymbol{\epsilon}_t\right)}{\sqrt{\alpha_t}} \right) + \sqrt{1-\alpha_{t-1}} \cdot \left( \boldsymbol{\epsilon}_t^{(\theta)}(\boldsymbol{x}_t) + \Delta\boldsymbol{\epsilon}_t \right) \tag{11}$$

$$= \sqrt{\alpha_{t-1}}\, \mathbf{P}_t(\boldsymbol{\epsilon}_t^\theta(\boldsymbol{x}_t)) + \mathbf{D}_t(\boldsymbol{\epsilon}_t^\theta(\boldsymbol{x}_t)) - \frac{\sqrt{\alpha_{t-1}}\sqrt{1-\alpha_t}}{\sqrt{\alpha_t}} \cdot \Delta\boldsymbol{\epsilon}_t + \sqrt{1-\alpha_{t-1}} \cdot \Delta\boldsymbol{\epsilon}_t \tag{12}$$

$$= \boldsymbol{x}_{t-1} + \left( -\frac{\sqrt{1-\alpha_t}}{\sqrt{1-\beta_t}} + \sqrt{1-\alpha_{t-1}} \right) \cdot \Delta\boldsymbol{\epsilon}_t \tag{13}$$

$$= \boldsymbol{x}_{t-1} + \left( -\frac{\sqrt{1-\alpha_t}}{\sqrt{1-\beta_t}} + \frac{\sqrt{1 - \prod_{s=1}^{t-1}(1-\beta_s)}\sqrt{1-\beta_t}}{\sqrt{1-\beta_t}} \right) \cdot \Delta\boldsymbol{\epsilon}_t \tag{14}$$

$$= \boldsymbol{x}_{t-1} + \left( \frac{\sqrt{1-\alpha_t-\beta_t} - \sqrt{1-\alpha_t}}{\sqrt{1-\beta_t}} \right) \cdot \Delta\boldsymbol{\epsilon}_t \quad \because \alpha_t = \prod_{s=1}^t (1-\beta_s) \tag{15}$$

$$\therefore \Delta\boldsymbol{x}_t = \tilde{\boldsymbol{x}}_{t-1} - \boldsymbol{x}_{t-1} = \left( \frac{\sqrt{1-\alpha_t-\beta_t} - \sqrt{1-\alpha_t}}{\sqrt{1-\beta_t}} \right) \cdot \Delta\boldsymbol{\epsilon}_t \text{ is negligible} \quad \because \beta_t < \beta_{max} \tag{16}$$

$\square$

# D    ADDITIONAL SUPPORTS FOR *h-space* WITH ASYRP

## D.1    RANDOM PERTURBATION ON $\epsilon$-*space* WITHOUT ASYRP

In § 3.1, we argue that if both $\mathbf{P}_t$ and $\mathbf{D}_t$ are shifted, we can not manipulate $\boldsymbol{x}_0$. In Figure 13, we do not observe the noticeable difference between (a) and (b) which are the result of the original reverse process of DDIM and the one with shifting both terms, respectively.

## D.2    ROBUSTNESS AND SEMANTICS IN *h-space* AND $\epsilon$-*space* WITH ASYRP

In § 3.2, we also argue that *h-space* is more robust than $\epsilon$-space with Asyrp. In Figure 13, we observe that small random noise $z \sim \mathcal{N}(0, \mathbf{I})$ in $\epsilon$-space degrades the resulting image without semantic changes (c) and much larger random noise in *h-space* yields random semantic changes without severe artifacts (d).

Note that diffusion models are designed as latent variable models with learned Gaussian transitions and the reverse process should also be close to Gaussian. Based on the assumption, we manage $\epsilon_t^\theta$ to follow the original Gaussian distribution as follows. Adding $z \sim \mathcal{N}(0, \sigma\mathbf{I})$ to $\epsilon_t^\theta$ expands the distribution of the predicted noise and may produce distorted images. To preserve the distribution, we scale $\tilde{\epsilon}_t^\theta = (\epsilon_t^\theta + z)/\sqrt{1^2 + \sigma^2}$. Still, the resulting images are almost identical compared to the original images where $\tilde{\epsilon}_t^\theta = \epsilon_t^\theta + z$ as shown in Figure 13. It is no wonder that the scaling does not improve the distorted results since the additive random noise disturbs the denoising operation of the predicted random noise.

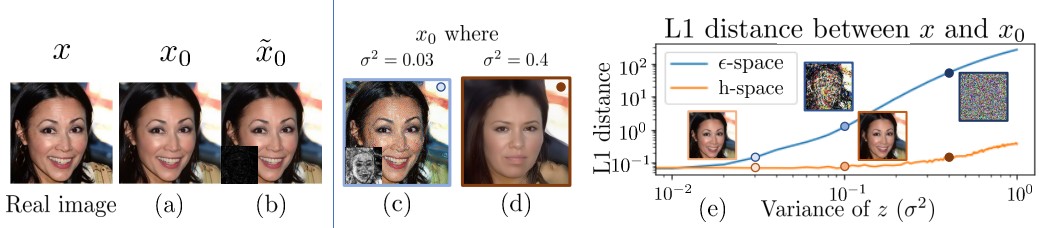

Figure 13: (a) The reconstructed image by the original DDIM inversion process which is almost indistinguishable to the real image. (b) The result from adding random noise $z$ both on $\mathbf{P}_t$ and $\mathbf{D}_t$. (a) looks identical to (b). The insets of (b, c) depict the full SSIM image from (a). It shows that simply shifting $\epsilon_t^\theta$ without Asyrp does not affect the result. (c) Adding $z \sim \mathcal{N}$ to $\epsilon$-space with Asyrp easily degrades image with little semantic change. (d) Adding $z \sim \mathcal{N}$ to *h*-space with Asyrp yields random semantic change without image degradation. (e) Correlation between image degradation and noise strength in the two spaces.

## D.3    CHOICE OF *h-space* IN U-NET

As shown in Figure 14, there are many other candidates for *h-space* in the architecture. Among the layers, we choose the 8th layer, the bridge of the U-Net based architecture. The layer is not influenced by any skip connection, has the smallest spatial dimension with compressed information, and is located just before the upsampling blocks. Thus, we assume that it could possibly be considered as the most suitable latent embedding. To confirm the assumption, we train $\boldsymbol{f}_t$ on the other layers. The results are shown in Figure 15. We carefully tuned the training hyperparameters ($\lambda_{CLIP}$ and $\lambda_{recon}$) for fair comparison. The 1st to the 6th layers hardly bring visible changes. The 7th and 9th layers bring not only the desired changes but also difficulty in finding optimal hyperparameters. After the 9th layer, the results bear severe artifacts.

# E    IMPLICIT NEURAL DIRECTIONS

Figure 16 illustrates the neural implicit function $\boldsymbol{f}_t$. It has only two 1x1 convolution layers with 512 channels. Note that we haven't explored the network architecture much.

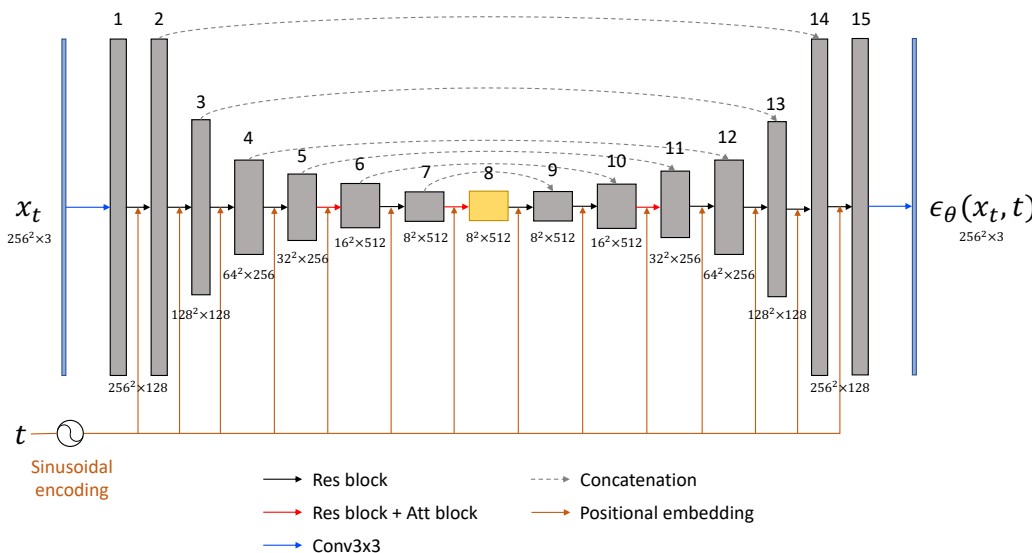

Figure 14: **Location of *h-space*.** The U-Net architecture of diffusion models outputs $256 \times 256$ images. Each layer is indexed with a number along the operating sequence of the model. The 8th layer is our *h-space* which is not directly influenced by a skip connection.

## F    QUALITY IMPROVEMENTS BY NON-ACCELERATED SAMPLING WITH SCALED $\Delta h_t$

Figure 17 shows the quality improvements by non-accelerated sampling with scaled $\Delta \boldsymbol{h}_t$ described in § 3.4. Even with different number of inference steps, we observe similar changes of an attribute if we preserve the sum of $\Delta \boldsymbol{h}_t$. This scaling technique allows non-accelerated sampling with 1000 steps for the models trained by accelerated training with 40 steps. Using non-accelerated sampling with scaled $\Delta \boldsymbol{h}_t$ leads to the same magnitude of manipulation and higher-quality images. In our experiments, it takes about 1.5 seconds to sample for 40 steps and 40 seconds for 1000 steps.

## G    EDITING STRENGTH AND EDITING FLEXIBILITY

### G.1    EDITING STRENGTH AND $t_{edit}$

Figure 18 shows the results according to $t_{\text{edit}}$. If $t_{\text{edit}}$ is too high, the length of the editing process becomes too short resulting in insufficient changes. On the contrary, too low $t_{\text{edit}}$ causes excessively unnecessary manipulation from the long editing process.

We observe that $t_{\text{edit}}$ is one of the important hyperparameters. We argue that the formula for choosing $t_{\text{edit}}$ using editing strength is reasonable because it applies to all five different datasets despite its sensitivity, even though the choice of LPIPS $= 0.33$ is empirical. We provide ablation of thresholds in Figure 22.

Additionally, these results imply why we need to use sufficiently low $t_{\text{edit}}$ in the unseen domains. Editing interval with insufficient editing strength struggles to escape from the training domain.

### G.2    EDITING FLEXIBILITY AND $t_{\text{boost}}$

Table 2 shows the prompts, $t_{\text{edit}}$, and $t_{\text{boost}}$. Intuitively, the attributes with larger visual changes have smaller cosine similarity and require longer editing interval. Figure 19 shows the average $\text{LPIPS}(\boldsymbol{x}, \mathbf{P}_t)$ and $\text{LPIPS}(\boldsymbol{x}, \boldsymbol{x}_t)$ of 100 samples on all datasets. Note that $t_{\text{edit}}$ and $t_{\text{boost}}$ dif-

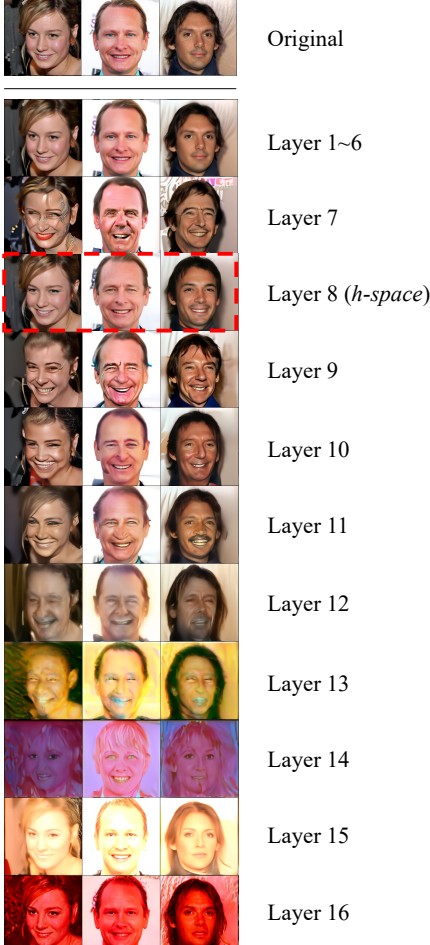

Original

Layer 1~6

Layer 7

Layer 8 (*h-space*)

Layer 9

Layer 10

Layer 11

Layer 12

Layer 13

Layer 14

Layer 15

Layer 16

Figure 15: **Exhaustive enumeration over the choices for semantic latent space.** We show the result of the training data. We observe that the eighth layer (*h-space*) of the U-net suits the best for the semantic latent space.

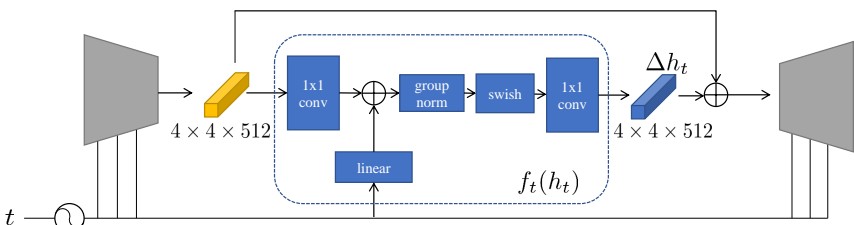

Figure 16: **Illustration of $f_t$.** We use group norm and swish following DDPM.

fer across datasets and attributes, and they are chosen by the formulas in § 4. We provide ablation of $t_{\mathrm{boost}}$ in Figure 23.

## H  QUALITY BOOSTING

We validate the effectiveness of our quality boosting (§ 4.2) in the original DDIM process and in Asyrp.

Smiling with

Real image           40 steps           1000 steps

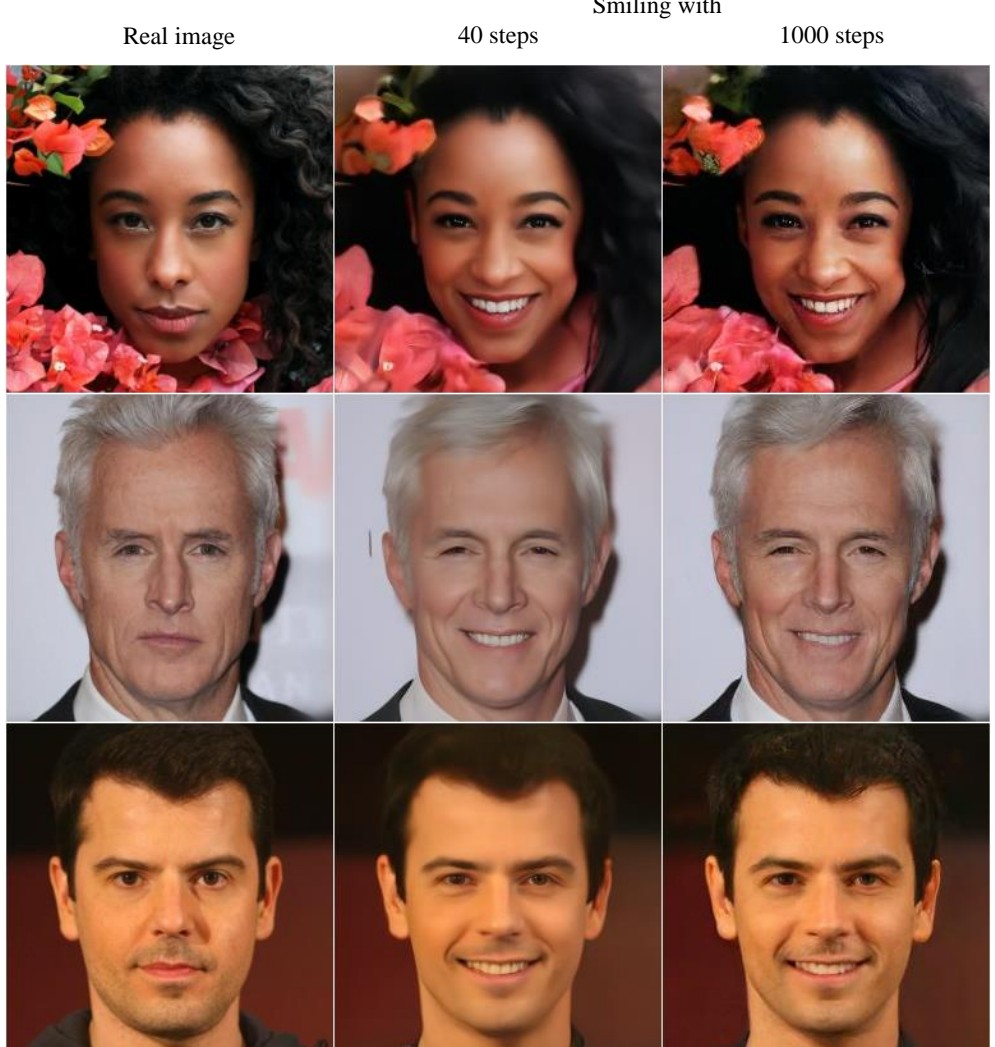

Figure 17: **Quality improvements by non-accelerated sampling with scaled** $\Delta \boldsymbol{h}_t$**.** We observe quality improvements by non-accelerated (1000-step) sampling with scaled $\Delta \boldsymbol{h}_t$ from accelerated (40-step) training described in § 3.4. Please zoom in for detailed comparison.

$t_{edit} = 900$   $t_{edit} = 800$   $t_{edit} = 700$   $t_{edit} = 600$   $t_{edit} = 500$   $t_{edit} = 400$   $t_{edit} = 300$   $t_{edit} = 200$   $t_{edit} = 100$

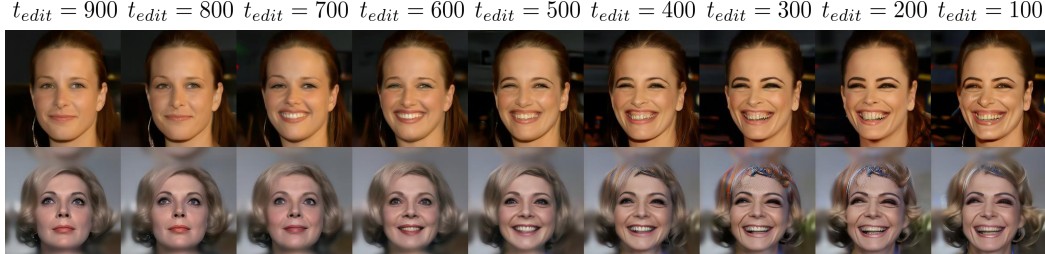

Figure 18: **Importance of choosing proper** $t_{\text{edit}}$**.** We explore various $t_{\text{edit}}$ with smiling. Too short editing interval struggles to manipulate attributes. Excessive editing strength results in degraded images.

| Dataset | src_txt | trg_txt | CLIP similarity | $t_{\text{edit}}$ | $t_{\text{boost}}$ |
|---------|---------|---------|-----------------|---------|---------|
| CelebA-HQ | Person | Young person | 0.905 | 515 | 167 |
| | Face | Smiling face | 0.899 | 513 | 167 |
| | Face | Sad face | 0.894 | 513 | 167 |
| | Face | Angry face | 0.892 | 512 | 167 |
| | Face | Tanned face | 0.886 | 512 | 167 |
| | Face | Disgusted face | 0.880 | 511 | 167 |
| | Person | Person with makeup | 0.875 | 509 | 167 |
| | Human | Zombie | 0.868 | 506 | 167 |
| | Person | Person with bald head | 0.861 | 505 | 167 |
| | Person | Person with curly hair | 0.835 | 499 | 167 |
| | Human | Neanderthal | 0.802 | 490 | 167 |
| | Person | Mark Zuckerberg | 0.797 | 489 | 167 |
| | Person | Nicolas Cage | 0.710 | 461 | 167 |
| | Human | Painting in the style of Pixar | 0.667 | 446 | 167 |
| | Photo | Painting in Modigliani style | 0.565 | 403 | 167 |
| | Photo | Self-portrait by Frida Kahlo | 0.443 | 321 | 167 |
| LSUN-church | Church | Gothic Church | 0.912 | 371 | 293 |
| | Church | Temple | 0.898 | 367 | 293 |
| | Church | Department store | 0.841 | 349 | 293 |
| | Church | Wooden House | 0.793 | 333 | 293 |
| | Church | Ancient traditional Asian tower | 0.784 | 330 | 293 |
| | Church | Red brick wall Church | 0.774 | 326 | 293 |
| | Church | Factory | 0.702 | 301 | 293 |
| LSUN-bedroom | Bedroom | Princess Bedroom | 0.912 | 370 | 221 |
| | Bedroom | Hotel Bedroom | 0.917 | 371 | 221 |
| AFHQ | Dog | Happy Dog | 0.883 | 430 | 167 |
| | Dog | Sleepy Dog | 0.866 | 422 | 167 |
| | Dog | Angry Dog | 0.860 | 419 | 167 |
| | Dog | Wolf | 0.850 | 412 | 167 |
| | Dog | Yorkshire Terrier | 0.690 | 302 | 167 |
| METFACES | Painting of a person | Painting of a Sad person | 0.921 | 330 | 180 |
| | Painting of a person | Painting of a Smiling person | 0.908 | 320 | 180 |
| | Painting of a person | Painting of a Disgusted person | 0.879 | 291 | 180 |

Table 2: Prompts, CLIP similarity, $t_{\text{edit}}$, and $t_{\text{boost}}$ for all attributes in the experiments.

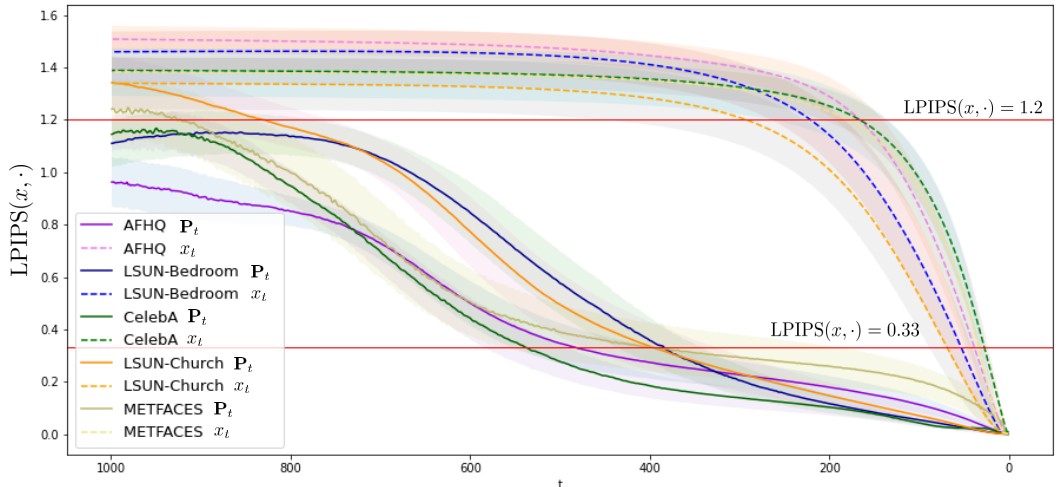

Figure 19: **Average** $\text{LPIPS}(\boldsymbol{x}, \mathbf{P}_t)$ **and** $\text{LPIPS}(\boldsymbol{x}, \boldsymbol{x}_t)$ **of 100 samples on all datasets.**

Figure 20 shows the effect of quality boosting with the original DDIM reverse process. Although the reverse process of DDIM has a nearly-perfect inversion property, we observe some noise by zooming in. Our quality boosting improves the quality of a sample and concurrently keeps nearly

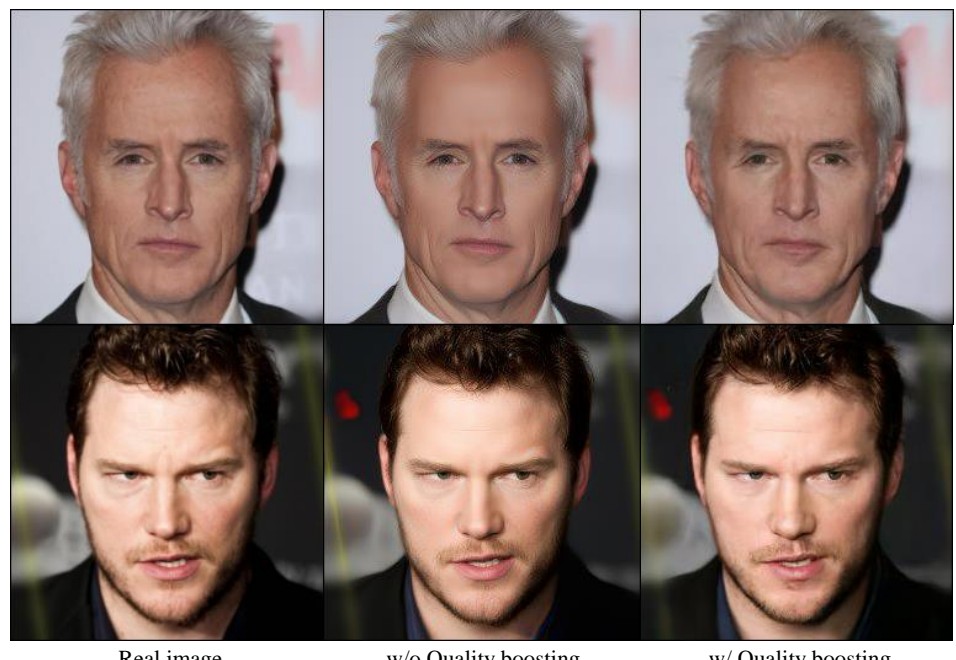

Figure 20: **Ablation study of quality boosting on the original DDIM process *without* Asyrp.** Our quality boosting enhances fine details and prevents images from being noisy in the original DDIM process. Please zoom in for detailed comparison.

Smiling

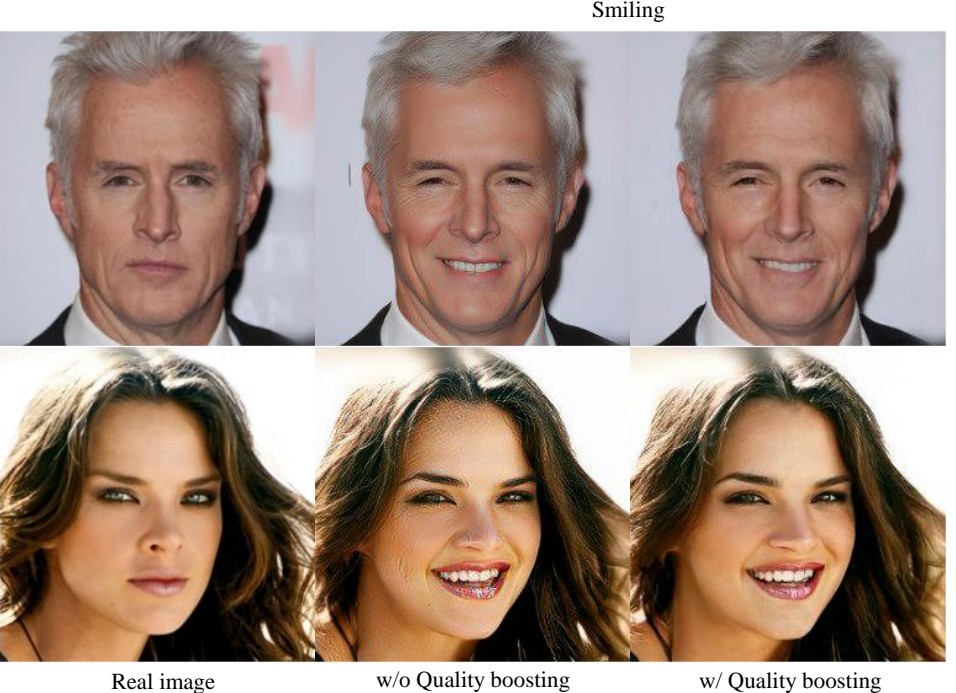

Figure 21: **Ablation study of quality boosting with Asyrp.** Our quality boosting enhances fine details and prevents images from being noisy. Note that the source of degradation is DDIM process, not Asyrp, confirmed in Figure 20. Please zoom in for detailed comparison.

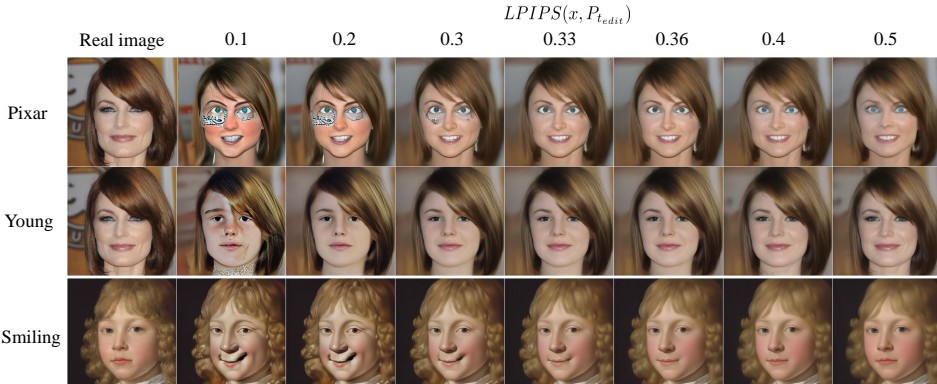

Figure 22: **Analyzing the effect of the hyperparameters** The figure shows that the calculated parameter works effectively as the maximum boundary of editing while maintaining the quality of the image.

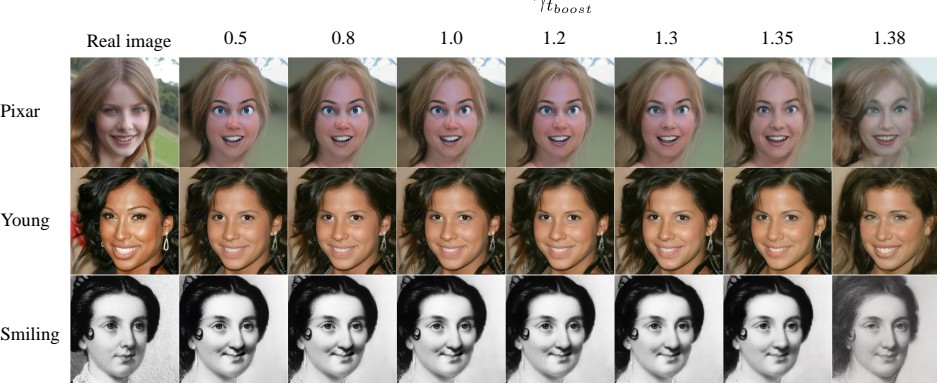

Figure 23: **Analyzing the effect of the hyperparameters** We observe that the quality of results has robustness to $\gamma_{t_{\mathrm{boost}}}$, except for too large $\gamma_{t_{\mathrm{boost}}}$.

perfect inversion property. We observe that $t_{\mathrm{boost}}$ is not sensitive, but the larger interval brings the less preservation.

Figure 21 shows quality improvements by our quality boosting in Asyrp.

# I ALGORITHM

Figure 24 illustrates generative process. Algorithm 1 and 2 describe training algorithm and inference algorithm of Asyrp, respectively.

# J TRAINING DETAILS

## J.1 LOSS COEFFICIENTS

Table 3 reports loss coefficients for each attribute. $\lambda_{\mathrm{CLIP}}$s near 0.8 are suitable for most in-domain attributes. For unseen domains, higher coefficient leads to more noticeable changes. Note that we use $\lambda_{\mathrm{recon}} = $ CLIP similarity $* 3$ which reduces L1 loss when an attribute needs a lot of change.

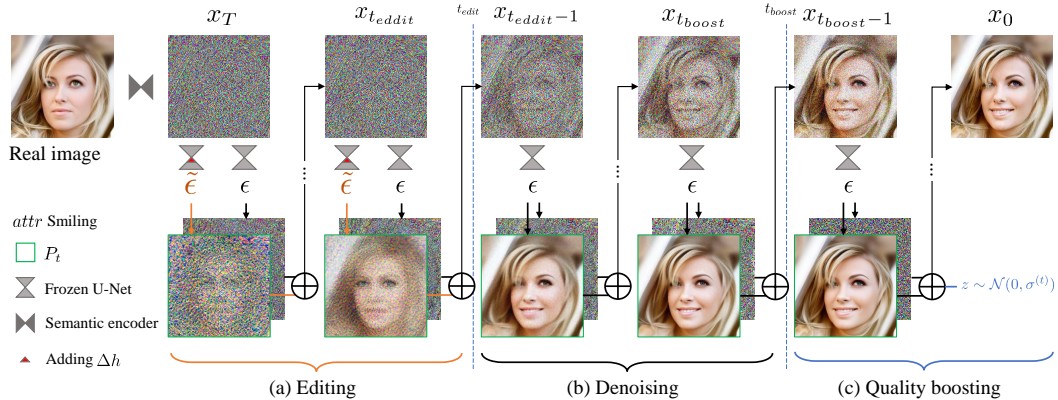

Figure 24: **Overview of our generative process.**

J.2    TRAINING WITH RANDOM SAMPLING INSTEAD OF THE TRAINING DATASETS

### J.2    TRAINING WITH RANDOM SAMPLING INSTEAD OF THE TRAINING DATASETS

Apparently, for training, inverting real-images can be replaced by random sampling. It refers to using $\boldsymbol{x}_T \sim \mathcal{N}(0, \mathbf{I})$ instead of $\boldsymbol{x}_T = q(\boldsymbol{x}_0) \cdot \prod_{t=1}^{T} q(\boldsymbol{x}_t \mid \boldsymbol{x}_{t-1})$ where $\boldsymbol{x}_0 \sim p_{data}(x)$. It allows us to train Asyrp only with the pretrained network and without extra dataset. Using random samples has tradeoff between preservation of contents and possible amount in editing. It take advantages when a target attribute requires large amount changes. We assume that the inversion of real-image is in the long tail of a Gaussian distribution because of realistic background or detailed clothes. On the other hand, random noise is considered to be closer to the mean of the normal, so it is easier to find directions. It can easily bring larger changes but also easily alter the contents. On the contrary, training with inversion shows the opposite property.

We train $\boldsymbol{f}_t$ with random sampling for attributes whose identity preservation is not important to take advantage of these properties. The rightmost column in Table 3 shows the choices.

## K    EVALUATION

### K.1    USER STUDY

We conduct user study to compare the performance of Asyrp and DiffusionCLIP (Kim & Ye, 2021) on Celeba-HQ (Liu et al., 2015) and LSUN-church (Yu et al., 2015). We use official checkpoints provided by DiffusionCLIP except for some facial attributes whose checkpoint do not exist. We tried our best to tune their hyperparameters following the manual for fair comparison. Example images are shown in Figure 25-27.

We use `smiling` and `sad` for in-domain CelebA-HQ attributes, `Pixar` and `Neanderthal` for unseen-domain CelebA-HQ attributes, and `department store`, `ancient`, and `wooden` for LSUN-church.

In unseen domain and Lsun-church, we use official checkpoints provided by DiffusionCLIP. We also randomly select 8 images for each CelebA-HQ attribute and 12 images for each LSUN-church attribute.

We observe that DiffusionCLIP works better in changing the holistic style of images. At the same time, it is short of the ability to bring semantic changes and suffers noisy results and a lack of diversity. The problems would be caused by fine-tuning the whole diffusion model.

We use the following questions for the survey. 1) Quality: Which image quality do you think is better? (clear and less noisy) 2) Attribute: Which image do you think is "Attribute(e.g., Smiling) naturally"? 3) Overall: Which image do you think is better considering the above evaluation criteria?

---

**Algorithm 1:** Editing(Inference)

---

**Input:** $x_0$(Input image), $\{f_t^i\}_{i=1}^M$(M Neural implicit functions for M attributes), $\{c_i\}_{i=1}^M$(user defined M scaling coefficients for M attributes), $\epsilon_\theta$ (frozen pretrained model), $S_{for}$ (# of inversion steps), $S_{gen}$(# of inference steps), $t_{\text{edit}}$ (computed from § 4.1), $t_{\text{boost}}$ (computed from § 4.2)

---

1 **Function** `Editor`$(x_0, \epsilon_\theta, \{c_i\}_{i=1}^M, S_{for}, S_{gen}, *)$:

    `// step 1: Semantic encoding`

2     Define $\{\tau_s\}_{s=1}^{S_{for}}$ s.t $\tau_1 = 0, \tau_{S_{for}} = T$

3     **for** $s = 1, 2, ..., S_{for} - 1$ **do**

4         $\epsilon \leftarrow \epsilon_\theta(x_{\tau_s}, \tau_s)$

5         $x_{\tau_{s+1}} = \sqrt{\alpha_{\tau_s}} x_{\tau_s} + \sqrt{1 - \alpha_{\tau_s}} \epsilon$

    `// step 2: Manipulation`

6     Define $\{\tilde{\tau}_s\}_{s=1}^{S_{gen}}$ s.t $\tilde{\tau}_1 = 0, \tilde{\tau}_{S_{edit}} = t_{edit}, \tilde{\tau}_{S_{noise}} = t_{\text{boost}}$ and $\tilde{\tau}_{S_{gen}} = T$

7     $\tilde{x}_{\tilde{\tau}_{S_{gen}}} = x_{\tilde{\tau}_{S_{for}}}$

8     **for** $s = S_{gen}, S_{gen} - 1, ..., 2$ **do**

        `// phase 1: editing`

9         **if** $s \geq S_{edit}$ **then**

10             Extract feature map $h_{\tilde{\tau}_s}$ from $\epsilon_\theta(\tilde{x}_{\tilde{\tau}_s})$

11             $\Delta h_{\tilde{\tau}_s} = \frac{S_{for}}{S_{gen}}(\sum_{i=1}^M c_i f_{\tilde{\tau}_s}^i(h_{\tilde{\tau}_s}))$

12             $\tilde{\epsilon} = \epsilon_\theta(\tilde{x}_{\tilde{\tau}_s} | \Delta h_{\tilde{\tau}_s})$

13             $\epsilon = \epsilon_\theta(\tilde{x}_{\tilde{\tau}_s})$

14             $\sigma_{\tilde{\tau}_s} = 0$

        `// phase 2: denoising`

15         **else if** $s \geq S_{noise}$ **then**

16             $\tilde{\epsilon} = \epsilon = \epsilon_\theta(\tilde{x}_{\tilde{\tau}_s})$

17             $\sigma_{\tilde{\tau}_s} = 0$

        `// phase 3: quality boosting`

18         **else**

19             $\tilde{\epsilon} = \epsilon = \epsilon_\theta(\tilde{x}_{\tilde{\tau}_s})$

20             $\sigma_{\tilde{\tau}_s} = \sqrt{(1 - \alpha_{\tilde{\tau}_{s-1}})/(1 - \alpha_{\tilde{\tau}_s})}\sqrt{1 - \alpha_{\tilde{\tau}_s}/\alpha_{\tilde{\tau}_{s-1}}}$

21         $z \sim N(0, 1)$

22         $\tilde{x}_{\tilde{\tau}_{s-1}} = \sqrt{\alpha_{\tilde{\tau}_{s-1}}}(\frac{\tilde{x}_{\tilde{\tau}_s} - \sqrt{1-\alpha_{\tilde{\tau}_s}}\tilde{\epsilon}}{\sqrt{\alpha_{\tilde{\tau}_s}}}) + \sqrt{1 - \alpha_{\tilde{\tau}_{s-1}} - \sigma_{\tilde{\tau}_s}^2}\epsilon + \sigma_{\tilde{\tau}_s}z$

23     **return** $\tilde{x}_0$ (manipulated image)

---

As for LSUN-church, we provide a set of four images at once and add a question: 3) Diversity: Which group do you think has a more diverse style? 4) Overall: Which image do you think is better considering the above evaluation criteria?

### K.2 SEGMENTATION CONSISTENCY AND DIRECTIONAL CLIP SIMILARITY

We compare Asyrp and DiffusionCLIP using directional CLIP similarity ($S_{\text{dir}}$) and segmentation-consistency (SC) following the protocols in DiffusionCLIP in Table 4 and Table 5. A pretrained CLIP (Radford et al., 2021) and segmentation models (Yu et al. (2018); Zhou et al. (2019; 2017); Lee et al. (2020)) are used to compute $S_{\text{dir}}$ and SC, respectively. We choose three attributes (`smiling`, `sad`, `tanned`) for CelebA-HQ-in-domain, two attributes (`Pixar`, `Neaderthal`) for CelebA-HQ-unseen-domain and three attributes (`department store`, `ancient`, `red brick`) for LSUN-church.

For a fair comparison, we use official checkpoints of DiffusionCLIP and provide scores of the attributes (`tanned`, `red brick`) following Kim & Ye (2021). Regarding attributes without the official checkpoints (`smiling`, `sad`), we train DiffusionCLIP by ourselves with the official code.

---

**Algorithm 2:** Training Neural implicit function $f_t$

---

**Input:** $\epsilon_\theta$(pretrained model), $\{x_0^{(i)}\}_{i=1}^N$(images to precompute), $f_t$(Neural implicit function of an attribute), $y_{src}$(source text), $y_{tar}$(target text), $S_{for}$(# of inversion steps), $K$(# of training epochs), $t_{\text{edit}}$ (computed from § 4.1), $t_{\text{boost}}$ (computed from § 4.2)

```
// step 1: Precompute latents
```

1 Define $\{\tau_s\}_{s=1}^{S_{for}}$ s.t $\tau_1 = 0, \tau_{S_{for}} = T$

2 **for** $i = 1, 2, ..., N$ **do**

3      **for** $s = 1, 2, ..., S_{for} - 1$ **do**

4          $\epsilon \leftarrow \epsilon_\theta(x_{\tau_s}^{(i)}, \tau_s)$

5          $x_{\tau_{s+1}}^{(i)} = \sqrt{\alpha_{\tau_s}} x_{\tau_s}^{(i)} + \sqrt{1 - \alpha_{\tau_s}}\epsilon$

6      Save the latent $x_{\tau_{S_{for}}}^{(i)}$

7 $\tau_{S_{edit}} = t_{edit}$

```
// step 2: Update  ft
```

8 **for** $epoch = 1, 2, ..., K$ **do**

9      **for** $i = 1, 2, ..., N$ **do**

10          Clone the latent $\tilde{x}_{\tau_{S_{for}}}^{(i)} = x_{\tau_{S_{for}}}^{(i)}$

11          **for** $s = S_{for}, S_{for} - 1, ..., S_{edit}$ **do**

12              Extract feature map $h_{\tau_s}$ from $\epsilon_\theta(\tilde{x}_{\tau_s}^{(i)})$

13              $\Delta h_{\tau_s} = f_{\tau_s}(h_{\tau_s})$

14              $P = \frac{\tilde{x}_{\tau_s}^{(i)} - \sqrt{1-\alpha_{\tau_s}}\epsilon_\theta(\tilde{x}_{\tau_s}^{(i)}|\Delta h_{\tau_s})}{\sqrt{\alpha_{\tau_s}}}; P_{src} = \frac{x_{\tau_s}^{(i)} - \sqrt{1-\alpha_{\tau_s}}\epsilon_\theta(x_{\tau_s}^{(i)})}{\sqrt{\alpha_{\tau_s}}}$

15              $\tilde{x}_{\tau_{s-1}}^{(i)} = \sqrt{\alpha_{\tau_{s-1}}}P + \sqrt{1 - \alpha_{\tau_{s-1}}}\epsilon_\theta(\tilde{x}_{\tau_s}^{(i)})$

16              $x_{\tau_{s-1}}^{(i)} = \sqrt{\alpha_{\tau_{s-1}}}P_{src} + \sqrt{1 - \alpha_{\tau_{s-1}}}\epsilon_\theta(x_{\tau_s}^{(i)})$

17              $L_{total} \leftarrow \lambda_{CLIP}L_{direction}(P, y_{tar}, P_{src}, y_{src}) + \lambda_{recon}|P - P_{src}|$

18              Take a gradient step on $L_{total}$ and update $f_t$

---

We use 100 samples per attribute. Asyrp outperforms DiffusionCLIP on $S_{\text{dir}}$ for all attributes. On SC, DiffusionCLIP achieves better or competitive scores. Because DiffusionCLIP manipulates images mostly by focusing on texture or color while preserving structure and shape, it takes advantage of getting higher SC scores. However, in Figure 28, results of `smiling` show that it is not proper to edit attributes which require structural manipulation. Note that better SC scores do not guarantee better qualitative performance. Results of `Pixar` also show the similar tendency of each method. We allow more structural changes than DiffusionCLIP while editing. Lower SC of our method comes from *desirable* structural changes as shown in Figure 28.

$$\mathcal{S}_{\text{dir}}\left(\boldsymbol{x}_{\text{gen}}, y_{\text{tar}}; \boldsymbol{x}_{\text{ref}}, y_{\text{ref}}\right) := \frac{\langle \Delta I, \Delta T \rangle}{\|\Delta I\|\|\Delta T\|}, \tag{17}$$

## L  DIRECTIONS

### L.1  GLOBAL DIRECTION

Figure 29 and Figure 30 show that the effects of mean direction and global direction are quite similar with $\Delta h_t$ by $f_t$ in various attributes. We compute mean direction and global direction from 20 different images.

We argue that *h-space* is roughly homogeneous across samples and timesteps. However, we observe that *h-space* is not completely independent to the conditions especially on unseen-domain

| Dataset | src_txt | trg_txt | $\lambda_{\mathrm{CLIP}}$ | $\lambda_{\mathrm{recon}}$ | from random noise |
|---|---|---|---|---|---|
| | Person | Young person | 0.8 | 0.905*3 | X |
| | Face | Smiling face | 0.8 | 0.899*3 | X |
| | Face | Sad face | 0.8 | 0.894*3 | X |
| | Face | Angry face | 0.8 | 0.892*3 | X |
| | Face | Tanned face | 0.8 | 0.886*3 | X |
| | Face | Disgusted face | 0.8 | 0.880*3 | X |
| | Person | Person with makeup | 0.8 | 0.875*3 | X |
| CelebA-HQ | Person | Person with curly hair | 0.8 | 0.835*3 | X |
| | Photo | Self-portrait by Frida Kahlo | 0.5 | 0.4433 | O |
| | Human | Zombie | 0.6 | 0.868*3 | O |
| | Person | Nicolas Cage | 0.8 | 0.710*3 | O |
| | Human | Painting in the style of Pixar | 0.8 | 0.667*3 | O |
| | Photo | Painting in Modigliani style | 0.8 | 0.565*3 | O |
| | Human | Neanderthal | 1.2 | 0.802*3 | O |
| | Church | Gothic church | 0.8 | 0.912*3 | O |
| | Church | Temple | 0.8 | 0.898*3 | O |
| | Church | Department store | 0.8 | 0.841*3 | O |
| LSUN-church | Church | Wooden house | 0.8 | 0.793*3 | O |
| | Church | Ancient traditional Asian tower | 0.8 | 0.784*3 | O |
| | Church | Red brick wall Church | 0.8 | 0.774*3 | O |
| | Church | Factory | 0.8 | 0.702*3 | O |
| LSUN-bedroom | Bedroom | Princess bedroom | 1.5 | 0.912*3 | O |
| | Bedroom | Hotel bedroom | 1.5 | 0.917*3 | O |
| | Dog | Happy dog | 1.5 | 0.883*3 | O |
| AFHQ-dog | Dog | Sleepy dog | 1.5 | 0.866*3 | O |
| | Dog | Wolf | 1.7 | 0.850*3 | O |
| | Dog | Yorkshire terrier | 2 | 0.690*3 | O |
| | Painting of a person | Painting of a Sad person | 1.5 | 0.921*3 | X |
| METFACES | Painting of a person | Painting of a Smiling person | 1.5 | 0.908*3 | X |
| | Painting of a person | Painting of a Disgusted person | 1.5 | 0.879*3 | X |

Table 3: The coefficients range from 0.5 to 0.8 for in-domain attributes. Unseen domains need slightly stronger $\lambda_{\mathrm{CLIP}}$s. We also report which attributes we train with random noise sampling. The criterion is which attributes require relatively less maintenance of identity. We use $\lambda_{\mathrm{recon}} =$ CLIP similarity $* 3$ which reduces L1 loss when an attribute needs a lot of change.

| | CelebA-HQ-in-domain | | | | | | CelebA-HQ-unseen-domain | | | |
|---|---|---|---|---|---|---|---|---|---|---|
| | Smiling | | Sad | | Tanned | | Pixar | | Neanderthal | |
| | $S_{\mathrm{dir}}$ | SC | $S_{\mathrm{dir}}$ | SC | $S_{\mathrm{dir}}$ | SC | $S_{\mathrm{dir}}$ | SC | $S_{\mathrm{dir}}$ | SC |
| Asyrp (ours) | **0.921** | 89.02% | **0.964** | 88.90% | **0.991** | 85.71% | **0.956** | 76.51% | **0.805** | 79.03 |
| DiffusionCLIP | 0.813 | **91.41%** | 0.760 | **89.93%** | 0.888 | **92.85%** | 0.811 | **89.91%** | 0.661 | **81.23%** |

Table 4: **Quantitative evaluation on CelebA-HQ.**

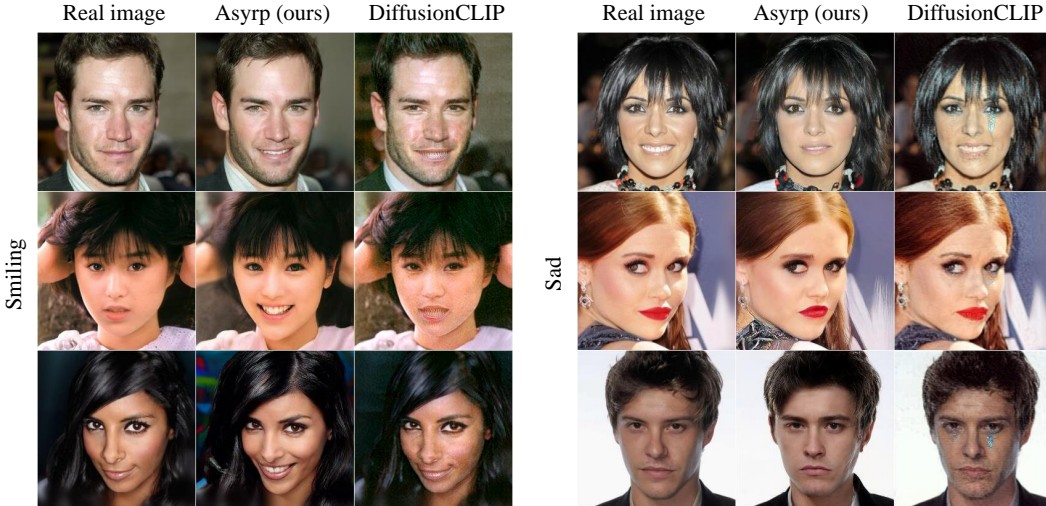

Figure 25: **Asyrp vs. DiffusionCLIP on CelebA-HQ in-domain attributes.** We observe that DiffusionCLIP struggles to change semantic facial attributes. Their official checkpoints do not exist and we ran careful hyperparameter tuning for training.

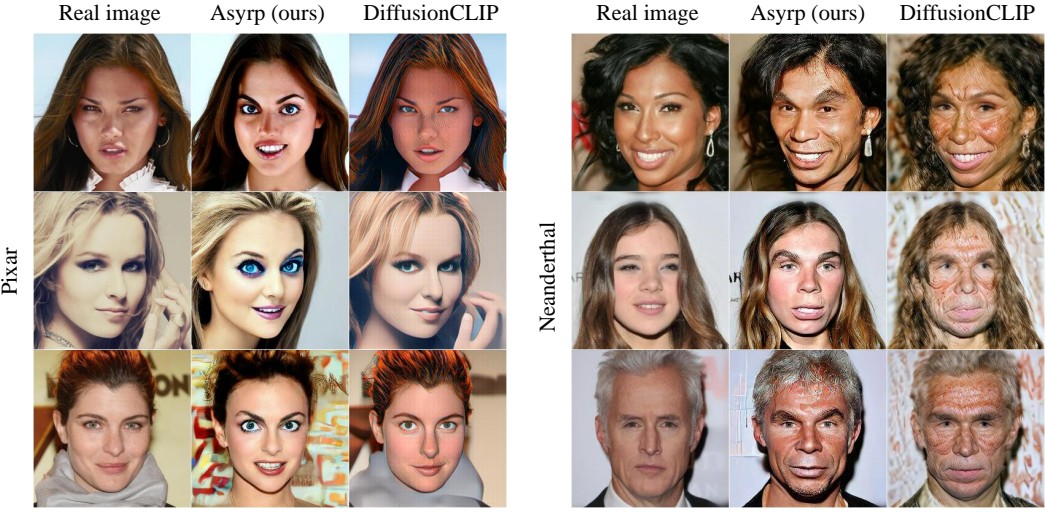

Figure 26: **Asyrp vs. DiffusionCLIP on CelebA-HQ unseen-domain attributes.** We use the official checkpoint provided by DiffusionCLIP. Asyrp works well even for the unseen-domains.

| | LSUN_church | | | | | |
| | Department store | | ancient | | red brick | |
| | $S_{\mathrm{dir}}$ | SC | $S_{\mathrm{dir}}$ | SC | $S_{\mathrm{dir}}$ | SC |
|---|---|---|---|---|---|---|
| Asyrp (ours) | **0.778** | **57.62%** | **0.943** | 62.65% | **0.989** | **65.83%** |
| DIffusionCLIP | 0.661 | 54.50% | 0.907 | **64.82%** | 0.964 | 65.02% |

Table 5: **Quantitative evaluation on LSUN-church.**

(See Figure 29). Note that unseen-domains require a longer editing interval with small $t_{\mathrm{edit}}$. Therefore, we conjecture that the consistency of *h-space* decreases at the end of the generative process. It is supported by additional experiments that L2 distance between $\Delta h_t$ and global direction gradually increases along with timesteps. We leave a more detailed analysis on *h-space* at different timesteps as a future work.

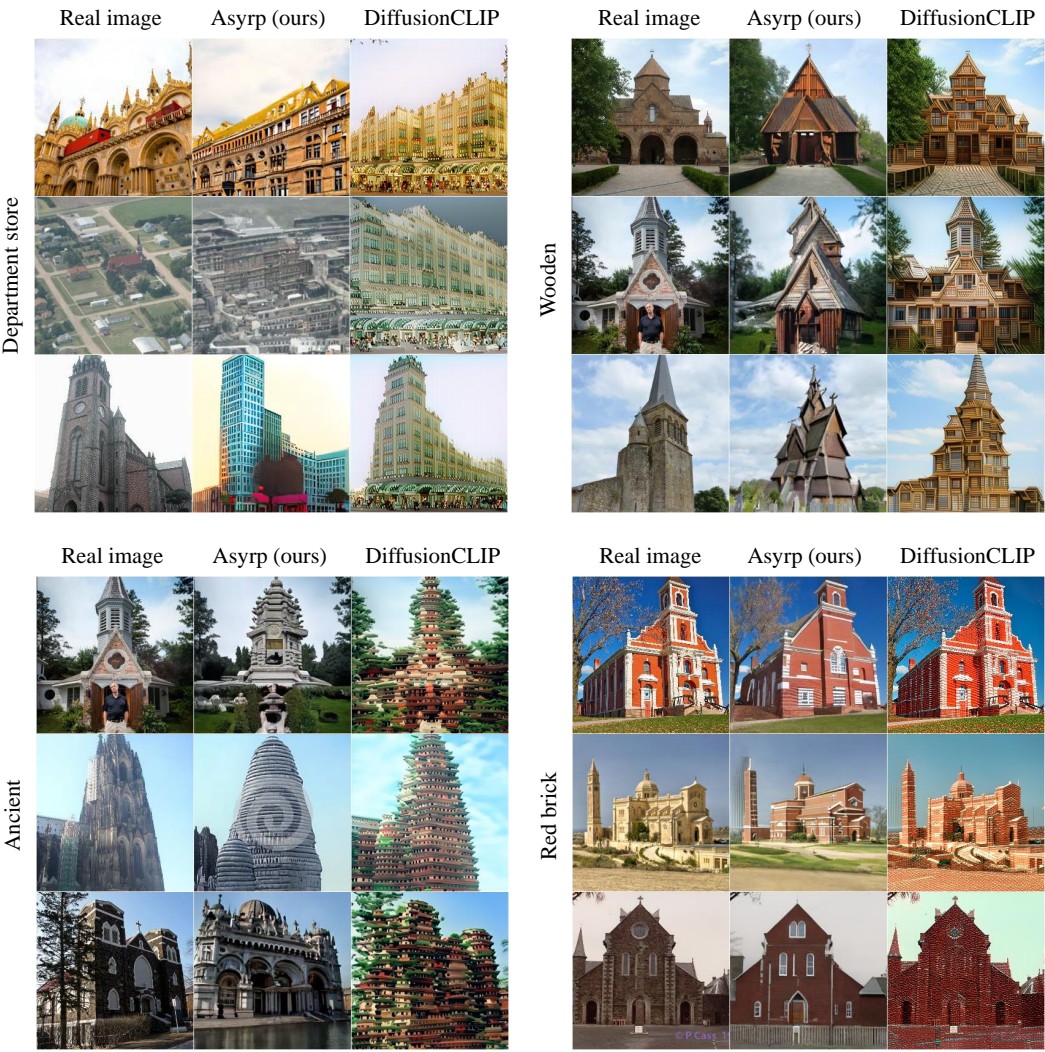

Figure 27: **Asyrp vs. DiffusionCLIP on LSUN-church.** Note that we use the official checkpoint provided by DiffusionCLIP. We observe DiffusionCLIP produces narrow range of styles while Asyrp produces diverse styles.

## L.2   COMPARE THREE METHODS

In this section, we compare three methods: implicit neural direction $f_t$, optimized $\Delta h_t$, and optimized $\Delta h^{global}$.

**Training time**   $f_t \approx \Delta h^{global} < \Delta h_t$

We have to optimize each $\Delta h_t$ for each time step $t$. Additionally, it needs specific hyperparameters for each $\Delta h_t$, e.g., higher learning rates for larger $t$. On the contrary, time-consuming for $f_t$ is similar to optimizing $\Delta h^{global}$.

**Quality**   $f_t \approx \Delta h_t > \Delta h^{global}$

$f_t$ and $\Delta h_t$, where directions can be obtained for each timestep, have the best quality. As can be shown in Figure 10, $\Delta h^{global}$ is sometimes accompanied by slight differences in hair, etc.

**Extensibility**   $f_t > \Delta h_t > \Delta h^{global}$

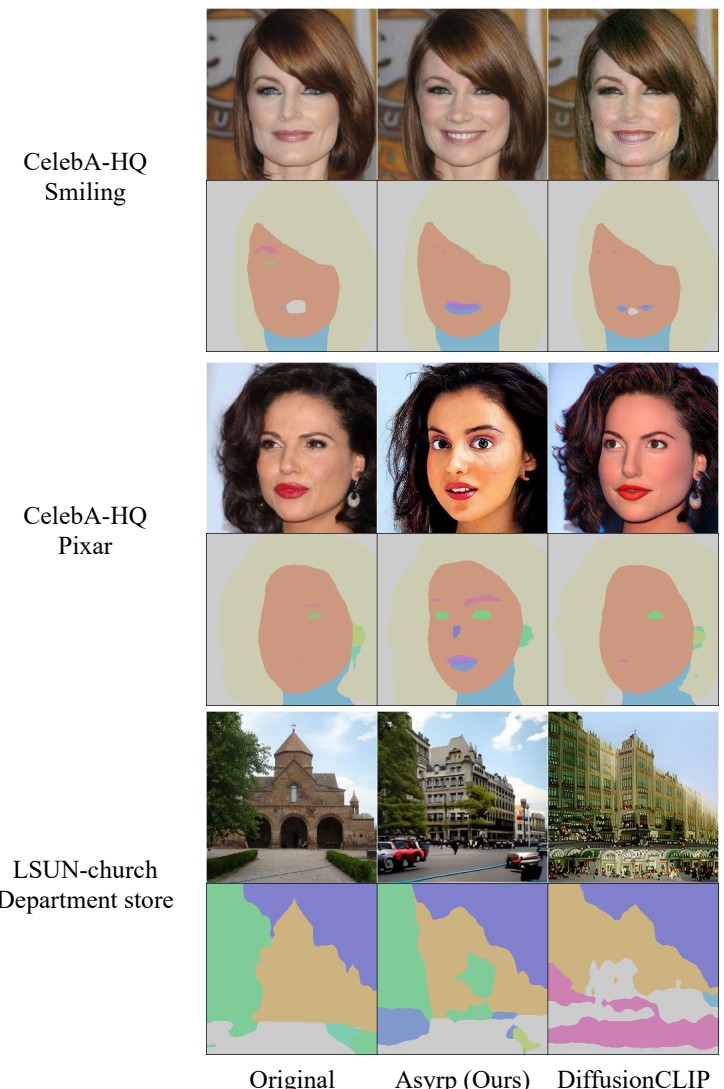

CelebA-HQ
Smiling

CelebA-HQ
Pixar

LSUN-church
Department store

Original     Asyrp (Ours)     DiffusionCLIP

Figure 28: **Example segmentations for computing segmentation-consistency (SC).**

$\Delta h_t$ can be obtained from $f_t$, and $\Delta h^{global}$ can be obtained by aggregating $\Delta h_t$.

We opt to use $f_t$ for above three advantages.

## M  RANDOM SAMPLING

We conduct extra experiments: generating images with target attributes using Asyrp not from inversion but from random Gaussian noises. As a consequence, the generative process can be used for conditional random sampling. We provide the results in Figure 31. However, it is beyond the scope of this paper.

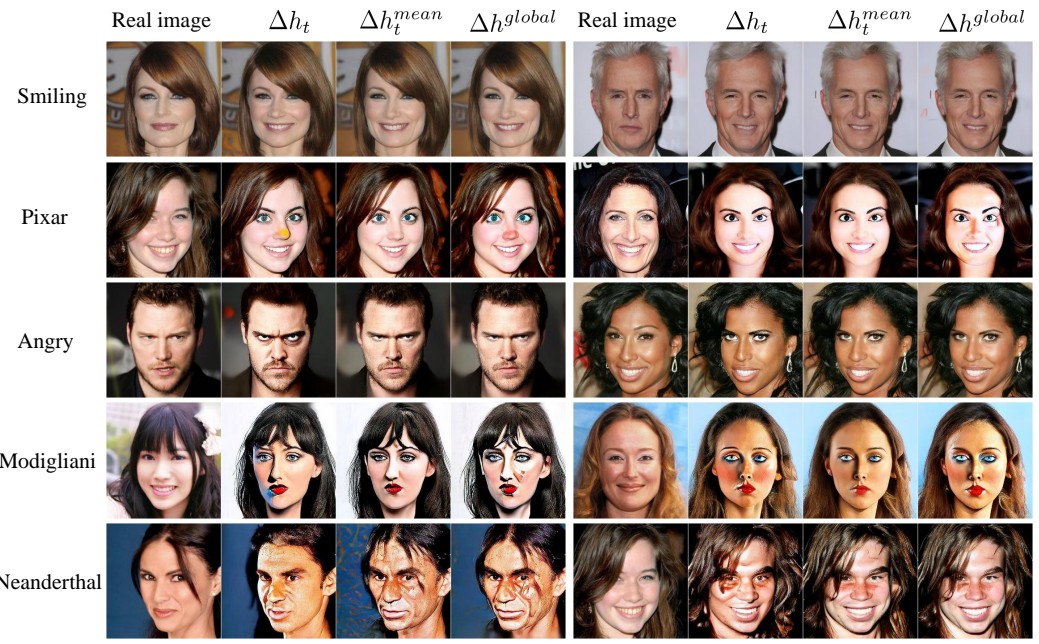

Figure 29: We compute mean direction and global direction from 20 other images on CelebA-HQ. The effect of mean direction and global direction are quite similar with $\Delta h_t$ by $f_t$ at diverse attributes.

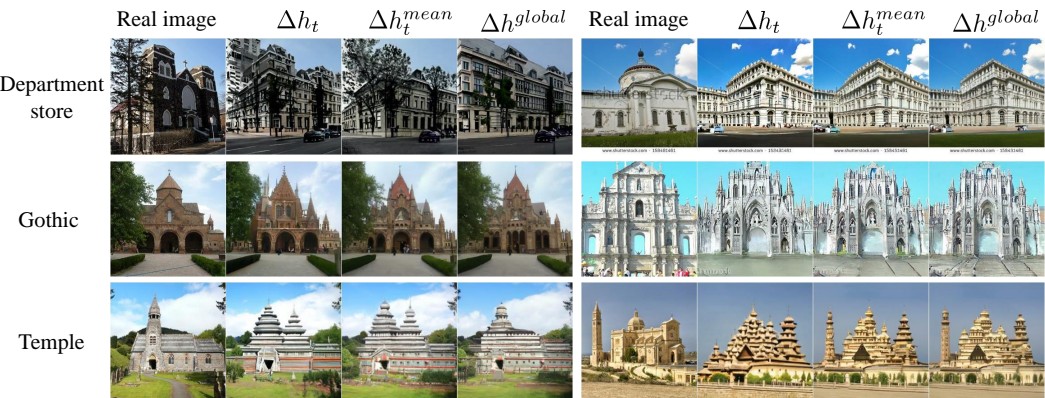

Figure 30: We obtain mean direction and global direction from 20 other images on LSUN-church. The effect of mean direction and global direction are quite similar with $\Delta h_t$ by $\boldsymbol{f}_t$ at diverse attributes.

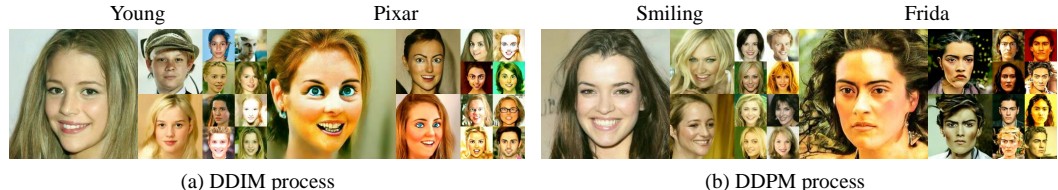

Figure 31: **Uncurated random sampling.** We generate images from random noise with Asyrp. Although we do not focus on these results, Asyrp can be used for conditional sampling.

# N    MORE SAMPLES

## N.1    IMAGENET

We conduct extra experiments: editing images with target class using Asyrp with ImageNet pre-trained model. We verified that models trained on large datasets, such as ImageNet, can be edited using Asyrp. However, we also observed that in this case the latent space is not partitioned by classes. For an orange, we have different latents for a single orange, for many oranges, for a cross-section of cut orange, and for a single piece of orange. Therefore, we learned the implicit function by collecting similar images to find the direction.

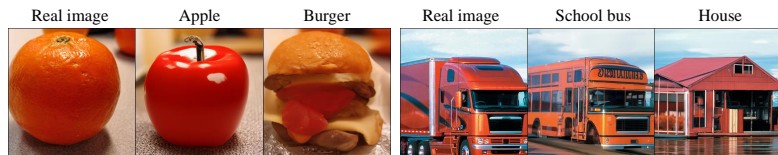

Figure 32: **Result of Asyrp in ImageNet.** The result shows that Asyrp works even in ImageNet dataset.

## N.2    MULTI-INTERPOLATION

Figure 33 provides mixed interpolation between multiple attributes. We observe that any interpolation with any attribute is possible.

## N.3    MORE RESULTS ON ALL DATASETS

We provide more results on CelebA-HQ (Figure 34), LSUN-church (Figure 35), AFHQ, LSUN-bedroom, METFACES (Figure 36).

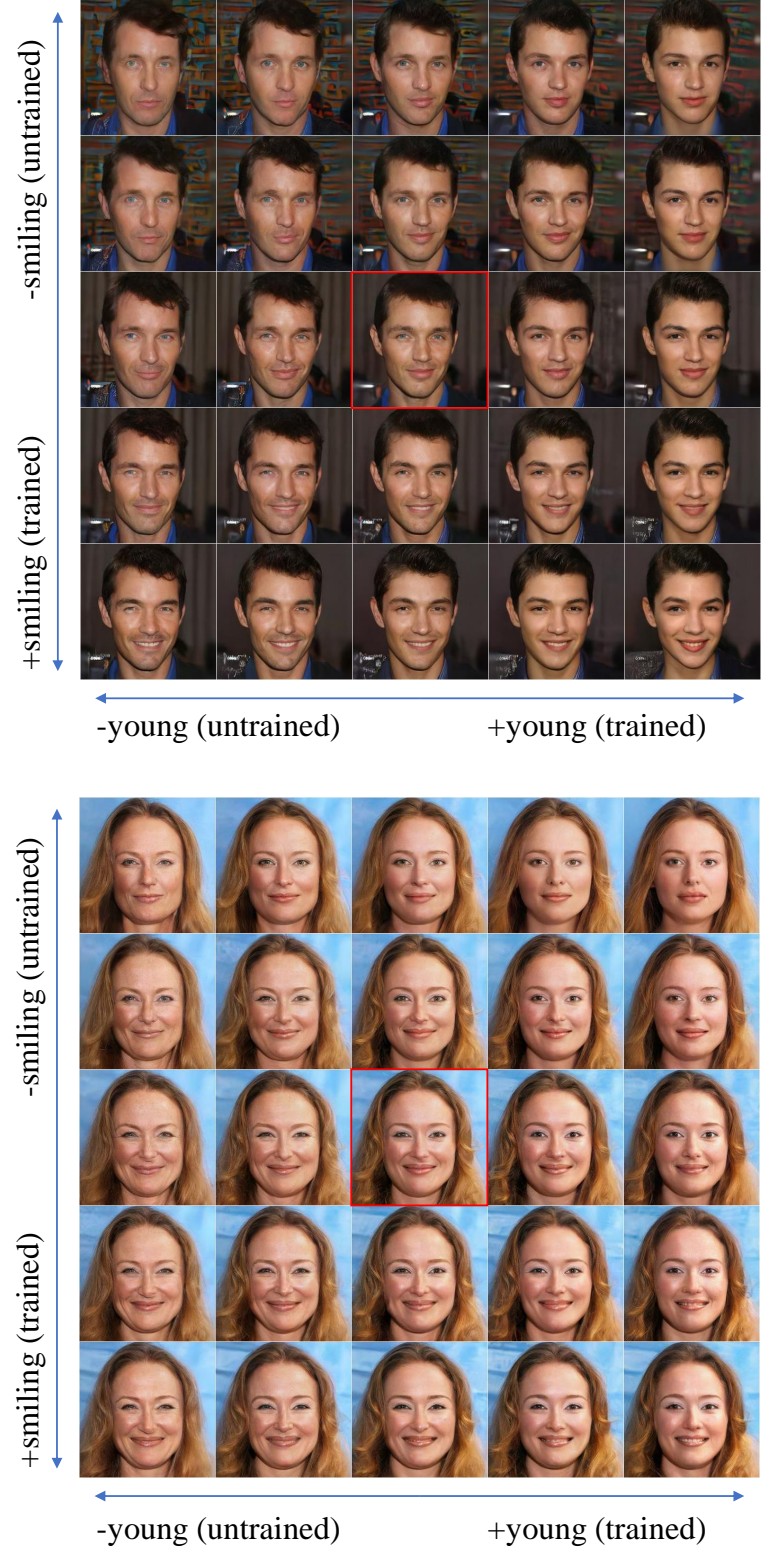

Figure 33: **Combination of multiple attributes.** The result shows that Asyrp works for combined $\Delta h$ of `smiling` and `young`.

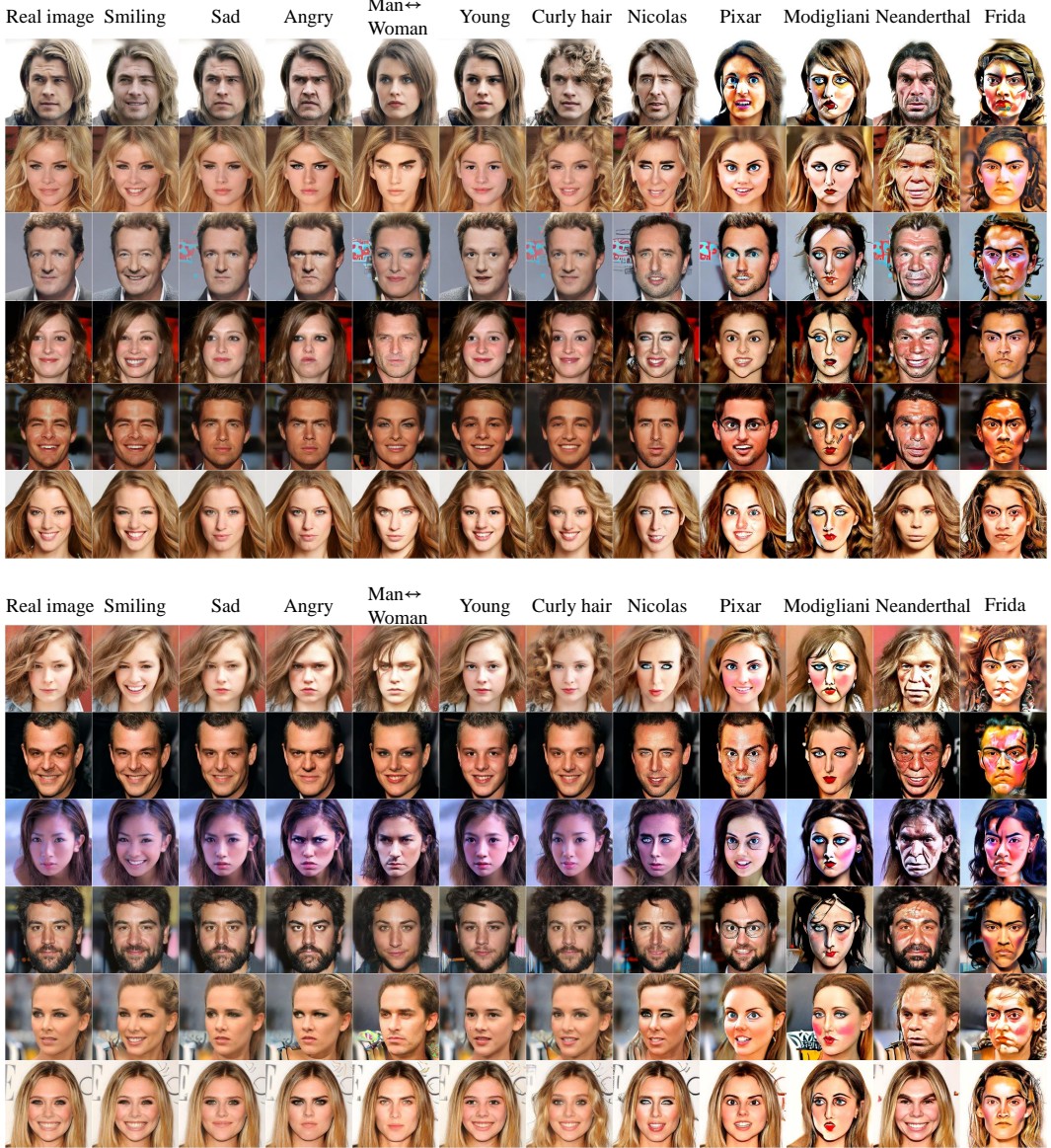

Figure 34: We provide more results on CelebA-HQ dataset.

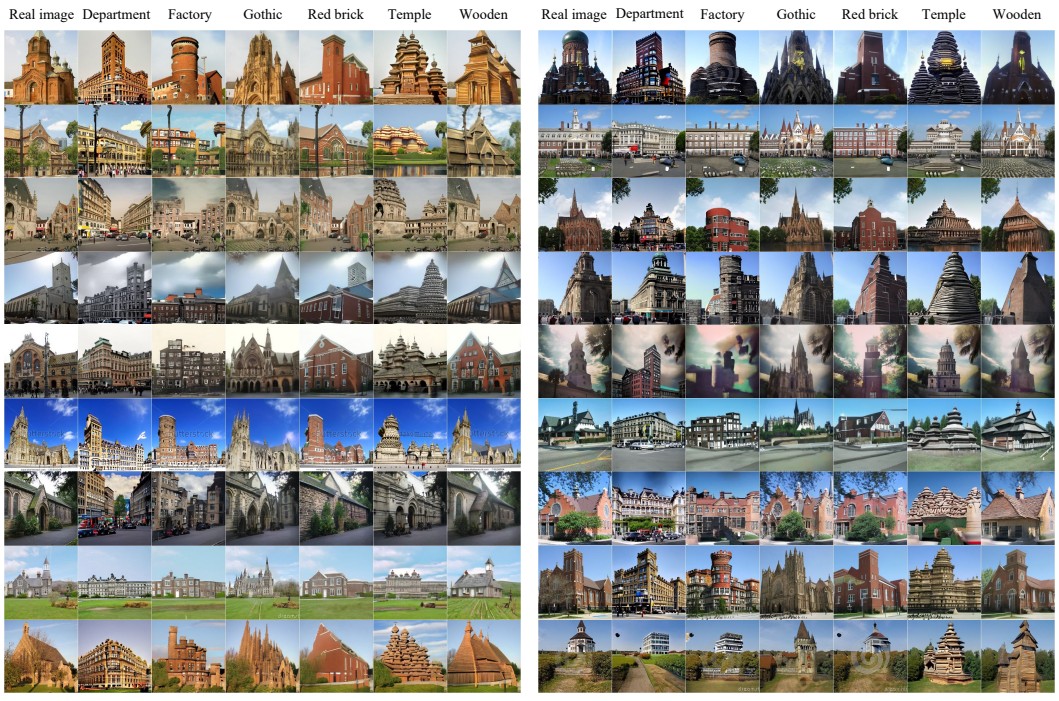

Figure 35: We provide more results on LSUN-church dataset.

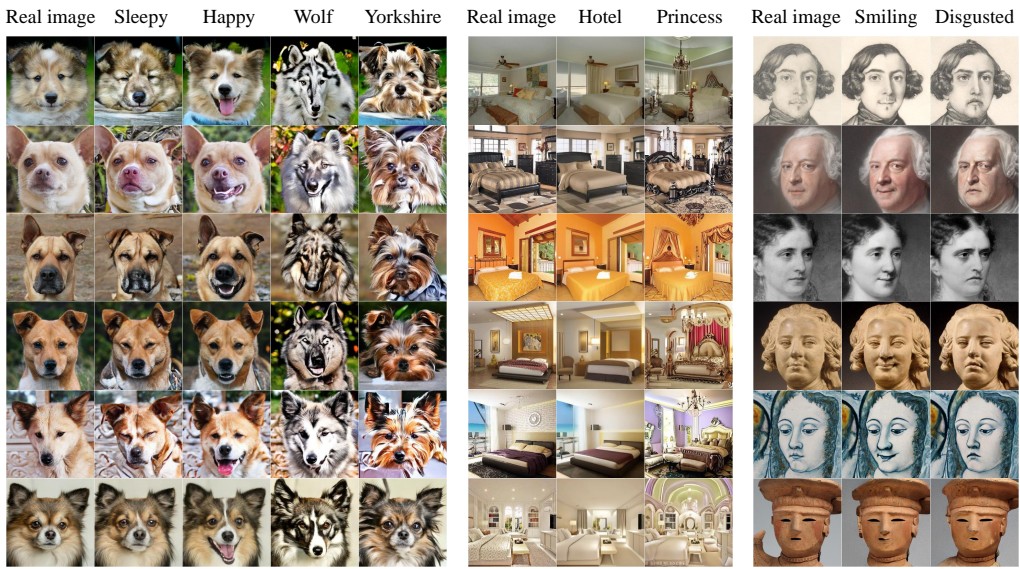

Figure 36: We provide more results on AFHQ, LSUN-bedroom and METFACES datasets respectively.

