# OpenReview forum: "Diffusion Models Already Have A Semantic Latent Space"
_ICLR.cc/2023/Conference — ICLR 2023 notable top 25%_

### Official Review · Reviewer_6SAm · 2022-10-19

**Confidence:** 4
**Correctness:** 3
**Technical Novelty And Significance:** 4
**Empirical Novelty And Significance:** 3
**Recommendation:** 6

**Clarity, Quality, Novelty And Reproducibility:**

I’m satisfied with the work's overall quality, clarity, and originality. There are some clarity-related questions:
- In Sec 3.2, I’m sure if it is reasonable to call $D_t$ the drift, which confuses me by thinking about the drift coefficient in the SDE formulation.
- In Sec 3.4, when we use a custom sub-sequence for the reverse process, why do we choose this normalization $S / \tilde{S}$?


**Strength And Weaknesses:**

Strengths:
- This work is clearly written and it is overall very easy to follow.
- I liked the idea of discovering the semantic directions in the h-space of existing diffusion models, since the state-of-the-art diffusion models use the UNet backbone. Compared with the noise space which has been considered in previous works, the h-space is more compact and contains higher semantic information. Moreover, the work semantically demonstrated in various experiments several nice properties of h-space: homogeneity, linearity, robustness and rough consistency across timesteps.
- Other important components in the proposed method, including asymmetric reverse process, implicit neural directions, three-stage generative process design, have also shown via experiments the respective effectiveness in contributing to the success of the proposed method.
- The experiments were conducted across different diffusion model architectures and different datasets to show the general applicability of the proposed method, and also demonstrated that it outperformed the previous fine-tuning method DiffusionCLIP.

Weaknesses:
- In the proof of Theorem 1, it assumed that $\beta_t \approx 0$. This might be true only when $t \to 0$. As $\beta_t$ monotonically increases from $t=0$ to $T$, I wonder if this assumption still holds when $t$ is large. I think more justifications are needed here to clarify the possible confusions.
- In experiments, many “less-diverse” datasets have been considered, such as CelebA-HQ, LSUN bedroom/church, AFHQ-dog. I wonder why not consider more diverse datasets, such as ImageNet? In particular, we know diffusion models have achieved good generation results on ImageNet. Compared with previous latent space traversal works based on StyleGAN that cannot work on ImageNet, this result will show a big advantage of the proposed method.
-  Since we already have many StyleGAN-based methods (StyleCLIP, StyleGAN-NADA, etc.) for latent semantic discovery, how does the proposed method based on diffusion models compare with them? I can tell that for the diffusion-model-based method, we still need many more tricks (e.g., asymmetric process, three-stage scheduling, implicit direction, etc.) to make it work reasonably well. I’m not saying that we have to beat those methods in this paper, but even not, it will still be interesting to see the pros and cons of both two methods and to provide guidelines for further improvements.


**Summary Of The Paper:**

This work proposed a new method called Asyrp (asymmetric reverse process) to discover the semantic directions in the latent space of pre-trained diffusion models. Specifically, this work first theoretically and empirically showed that a shift added to the UNet noise prediction results in almost the same reverse process trajectory. To this end, an asymmetric reverse process is proposed to manipulate the real image by controlling the latent variable. Besides, this work found that the bottleneck (or the deepest) feature map in the UNet, termed h-space, has the most salient properties than other latent spaces, and thus considered to learn implicit neural directions in the h-space. Finally, in order to achieve the best tradeoff between editing strength and generation quality, this work designed a full editing process, consisting of three phases: editing with Asyrp, traditional denoising, and quality boosting.


**Summary Of The Review:**

Overall, I liked the idea of discovering the latent semantics in diffusion models and I think the experiments well support the claimed contributions in the work. Since I still have some concerns about the theory and experiments (see the weaknesses above), I put my initial rating as “weak accept”. I’m open to increasing my score if the rebuttal well addresses my concerns.

---

> ### Author Response · Authors · 2022-11-09
> **Response to Reviewer 6SAm (2)**
>
> >[W3] Since we already have many StyleGAN-based methods (StyleCLIP, StyleGAN-NADA, etc.) for latent semantic discovery, how does the proposed method based on diffusion models compare with them? I can tell that for the diffusion-model-based method, we still need many more tricks (e.g., asymmetric process, three-stage scheduling, implicit direction, etc.) to make it work reasonably well. I’m not saying that we have to beat those methods in this paper, but even not, it will still be interesting to see the pros and cons of both two methods and to provide guidelines for further improvements.
>
> It is a very good discussion. We add it to Appendix B, More discussion. Thank you for your suggestion.
>
> >> In this section, we discuss the pros and cons of diffusion-model-based and GAN-based methods. And we provide guidelines for further improvements.
> GAN-based latent manipulation methods (StyleCLIP and StyleGAN-NADA) require careful inversion from real images to latent codes for real image editing. On the contrary, our proposed method based on diffusion models has a powerful advantage; the sophisticated inversion method is not necessary. This means that we can obtain the latent code of an arbitrary real image even if the image is not in the trained domain.
> On the other hand, several inversion methods have been proposed for GANs to obtain the latent of the real image, and the corresponding latent manipulation method should be considered for each inversion method. For example, it is difficult to apply the method of editing in $\mathcal{W}$ space to the method of inversion using $\mathcal{W}^+$ space.
> However, GANs have the advantage of fast sampling. In addition, diffusion models have a relatively slow sampling time. Additionally, we have to be aware of the time steps of diffusion models, which is still less well known.
> The advantage of being free from Inversion provides the following milestones. The manipulation in the latent of the diffusion models is the same as the editing in real images. It can be expanded to segmentation, clustering, classification, etc. in h-space for real-world images.
> It would be an interesting research direction to employ previous techniques. Our method can be used in conjunction with gradient guidance methods. Although we do not focus on random sampling, ours works effectively for sampling with stochastic. (See Appendix M. Random Sampling) It may bring more diverse methods to steer diffusion models.
> h-space in the latent diffusion models such as stable diffusion, is another interesting research direction. The main contribution of our paper is only modifying $P_t$ while preserving $D_t$, and can be adapted with latent diffusion models. However, since the latent meaning may be different due to structural differences, research on this is needed.
> Furthermore, all of the properties of h-space according to the time step has not been fully discussed so far. Research on them can be expected to expand further.
>
> ******
> > [clarity-related 1] In Sec 3.2, I’m sure if it is reasonable to call the drift, which confuses me by thinking about the drift coefficient in the SDE formulation.
>
> We modify the word "drift" in Sec 3.2. We agree that it can cause confusion even if we use literally "drift".
>
> - original drift $D_t$ $\to$ original flow $D_t$
> - the verb "drift" $\to$ "modify"
>
> These are marked in blue in the updated version.
>
> *****
> > In Sec 3.4, when we use a custom sub-sequence for the reverse process, why do we choose this normalization ?
>
> The $S/\tilde{S}$ is for maintaining the total amount of change. We added a sentence to help understanding.
> >>It preserves the amount of total change $\sum\Delta h_t$, according to $\Delta \tilde{h}_{\tilde{\tau}} \tilde{S}=\Delta h_t S$.
>
> Here we provide an example.
> Assuming that there is $\Delta h_t$ learned with 10 intervals. If we want to sample with 50 intervals, using the $\Delta h_t$ learned with 10 intervals would lead to 5x amount of change. So we multiply 10/50 to reduce the amount of change.
>
> ******
> Thank you for the constructive comments. We hope these explanations resolve the concerns. Further questions or suggestions would be also appreciated.

---

> ### Author Response · Authors · 2022-11-09
> **Response to Reviewer 6SAm (1)**
>
> Thank you for acknowledging our strengths:
> - asymmetric reverse process to make h-space as a semantic latent space of pre-trained diffusion models.
> - demonstrating in various experiments several nice properties of h-space.
> - showing the general applicability.
> *****
> > [W1] In the proof of Theorem 1, it assumed that  $\beta_t \approx 0$. This might be true only when  $t\to0$. As $\beta_t$ monotonically increases from $t=0$ to $T$, I wonder if this assumption still holds when $t$ is large. I think more justifications are needed here to clarify the possible confusions.
>
> First, when $t=T$, $\beta_T$ is defined as a samll value e.g., $\beta_T=0.001$. However it can be misleading, therefore we add more justifications and modify the proof of Theorem 1 as follows:
> (The modified part of below was marked with a bold. And the paper is marked in blue.)
>
>
> >> Theorem 1. Let $\epsilon_t^{\theta}$ be a predicted noise during the original reverse process at $t$ and $\tilde{\epsilon_{t}^{\theta}}$ be its shifted counterpart. **Then, $\Delta x_t=\tilde{x_{t-1}} - x_{t}$ is negligible** where $\tilde{x_{t-1}} = \sqrt{\alpha_{t-1}}\ P_t(\tilde{\epsilon}_t^{\theta}(x_t)) + D_t(\tilde{\epsilon}_t^{\theta}(x_t))$.
> I.e., the shifted terms of $\tilde{\epsilon}_t^{\theta}$ in $P_t$ and $D_t$ destruct each other in the reverse process.
>
> >> Proof. **Define $\tilde{\epsilon_t^{\theta}} (x_t)=\epsilon_t^{\theta}(x_t)+\Delta\epsilon_{t}$, {$\beta_{t}$}$ㅤ_{t=1}^{T}$={$\beta_{1}$ $=\beta_{min}$, ..., $\beta_{T}=\beta_{max}$}, and $\alpha_{t}=\prod_{s=1}^{t} (1-\beta_s)$. Note that $\beta_{max}$ is defined as a small value (e.g., $\beta_{max} = 0.001$) and {$\beta_{t}$}$ㅤ_{t=1}^{T}$ are defined by a decreasing schedule from $\beta_T= \beta_{max}$ to $\beta_1 = \beta_{min} \approx 0$ (e.g., $\beta_{min} = 0.00001$).** Then, ...
>
>
> *****
> > [W2] -   In experiments, many “less-diverse” datasets have been considered, such as CelebA-HQ, LSUN bedroom/church, AFHQ-dog. I wonder why not consider more diverse datasets, such as ImageNet? In particular, we know diffusion models have achieved good generation results on ImageNet. Compared with previous latent space traversal works based on StyleGAN that cannot work on ImageNet, this result will show a big advantage of the proposed method.
>
> Thank you for the interesting suggestion. We will conduct experiments on ImageNet to add the results and discussion in the camera ready. Please note that the coverage of latent editing on ImageNet is very narrow even in BigGAN, such as geometric transformations (translation, zoom) and a few semantic changes (apple->orange).
>
> *****

---

> ### Author Response · Authors · 2022-12-02
> **Response to Reviewer 6SAm**
>
> Again, thank you for your efforts and service. We would be appreciated it if you check out our revision of our paper according to your concerns. Please let us know if we have addressed all your concerns.
>
> The experiment on ImageNet is in process. We will let you know as soon as it is done.
> Thank you.

---

> ### Author Response · Authors · 2022-12-10
> **Response to Reviewer 6SAm**
>
> We provide a few example usages of Asryp on ImageNet. Following the anonymity policy, we provide the results through an anonymous link https://ibb.co/ChfZbwF. It demonstrates versatile editing capability of our method. We will add more images in the final version.
>
> Thank you.

---

> > ### Comment · Reviewer_6SAm · 2022-12-13
> > **Thank you**
> >
> > Thank you for providing detailed response. First of all, I apologize for a very late response to your rebuttal. Regarding my concerns, 1) *proof of Theorem 1*: I’m not sure how you set $\beta_{\text{max}}$ and $\beta_{\text{min}}$. But according to Score SDE [1], we have $\beta_{\text{max}}=0.02$ and $\beta_{\text{min}}=0.0001$. Other than that, other clarifications make sense to me. 2) *comparisons with GAN-based methods*: I appreciate the helpful discussions provided by the authors. On top of that, however, I think adding some direct experimental comparisons will make this paper stronger. 3) *new experiments on ImageNet*: I’m not sure how the results demonstrate the editing performance given limited information. Perhaps the authors wanted to show the CLIP-based editing? I hope to see more qualitative and quantitative results. All the above, I keep my rating unchanged.
> >
> > [1] Song et al., Score-Based Generative Modeling through Stochastic Differential Equations, ICLR 2021.

---

> > > ### Author Response · Authors · 2022-12-13
> > > **Thank you**
> > >
> > > Thank you for your response.
> > >
> > > 1. Following official training setting, we use $\beta_{\max}=0.02$ and $\beta_{\min}=0.0001$ for VP-SDE, iDDPM, ADM-G; and  $\beta_{\max}=0.001$ and $\beta_{\min}=0.00001$ for DDPM. We also empirically confirmed that $\Delta x_t$ is negligible in all possible configurations of $\beta$ (i.e., $\Delta x_t$ can't make a semantics change).
> > >
> > > 2. We omit the comparison to the GAN-based methods because DiffusionCLIP shows comparable performance and ours outperforms DiffusionCLIP. We will clarify this in the final version.
> > >
> > > 3. The additional ImageNet experiment shows that finding $\Delta h$ using directional CLIP objective generally works on various datasets including the ones in the paper (CelebA-HQ, AFHQ, LSUN-church/bedroom, and METFACES). We will add more qualitative and quantitative results following your constructive suggestions in the final version.
> > >
> > > Thank you again for the suggestions and the effort.

---

> > > > ### Comment · Reviewer_6SAm · 2022-12-13
> > > > **Follow up**
> > > >
> > > > Thank you for your prompt response. Now everything looks clear to me and I recommend an acceptance (assuming new results will be added in the final version).

---

### Official Review · Reviewer_KRCt · 2022-10-20

**Confidence:** 3
**Correctness:** 4
**Technical Novelty And Significance:** 4
**Empirical Novelty And Significance:** 3
**Recommendation:** 8

**Clarity, Quality, Novelty And Reproducibility:**

### Novelty
Traversing the latent space of a diffusion model causing semantic changes by manipulating the image representation is a novel contribution.

### Clarity
The procedure, though a bit convoluted, is clearly explained, and graphics support the understanding of the algorithm. Images in Figures 1 and 2 could be enlarged a bit to ease readability.

### Reproducibility
The algorithm is in pseudo-code in the appendix and the authors also provide code that will be published with their contribution. I have not checked the code, but if functional, reproducibility is provided.

### Quality
The results look great. The procedure is nicely explained. I also appreciate the background section 2.

### minor mistakes
There are some minor language mistakes. For example:
- in 2.1: that learns data distribution $\rightarrow$ that learns a data distribution
- in 2.1: is variance schedule $\rightarrow$ is the variance schedule
- in 3.1: given a pretrained and frozen diffusion models $\rightarrow$ given a pretrained and frozen diffusion model
- in 6: image editing on semantic latent space $\rightarrow$ image editing in the semantic latent space
- in A:  On the other hands $\rightarrow$ On the other hand
- in A:  incorporating diffusion models with scorebased model $\rightarrow$ incorporating diffusion models with scorebased models
- in A:  However it requires noise-dependent $\rightarrow$ However it requires a noise-dependent
- ...

**Strength And Weaknesses:**

### Strengths

The visual results are striking. There are plenty of examples of different attributes for editing. A user study (n=80) was conducted and showed very clearly superior performance to DiffusionCLIP. The algorithm can be applied to pretrained diffusion models.


### Weaknesses
Theorem 1 seems to be formulated a bit sloppy... $ \tilde{x}_t \approx  x_t $
isn't really saying anything. Defining a bound $||\tilde{x}_t -  x_t|| \leq \delta$ and then showing that $\delta$ is small would be better suited.
For the proof in appendix B, I would prefer if all symbols were defined in that section. In equation (13) you introduce $\beta$ and $\eta$ without stating their relation to previous terms.
In general, theoretical motivation for the procedure and parameter choices is lacking a bit.

The only quantitative evaluation is the user study. Results for DiffusionCLIP seem a bit lower quality than in the original paper.



**Summary Of The Paper:**

The authors introduce an algorithm that manipulates images using diffusion models.
They do so by manipulating the representation in the bottle-neck U-net layer of the diffusion model over several (but not all) timesteps. The loss function that they optimize is the same as for DiffusionCLIP.

**Summary Of The Review:**

Overall, I think this is a nice contribution that carefully explains the authors' procedure and provides visually striking experimental results. The contribution is novel and presents a significant step towards practical image editing with diffusion models. With additional theoretical justifications, this would be even better. I would recommend it for publication at ICLR.

---

> ### Author Response · Authors · 2022-11-09
> **Response to Reviewer KRCt**
>
> Thank you for acknowledging our main contributions:
> - the striking visual results and superior performance.
> - asymmetric reverse process to make h-space as a semantic latent space of pre-trained diffusion models.
>
> *****
> > [W1] Theorem 1 seems to be formulated a bit sloppy... $\tilde{x}_t \approx x_t$ isn't really saying anything. Defining a bound ||x~t−xt||≤δ and then showing that δ is small would be better suited. For the proof in appendix B, I would prefer if all symbols were defined in that section. In equation (13) you introduce β and η without stating their relation to previous terms. In general, theoretical motivation for the procedure and parameter choices is lacking a bit.
>
> We love your constructive suggestion.
> We added more justifications and modified the proof of Theorem 1 as follows:
> (The modified part of below was marked with a bold. And the paper is marked in blue.)
>
>
> >> Theorem 1. Let $\epsilon_t^{\theta}$ be a predicted noise during the original reverse process at $t$ and $\tilde{\epsilon_{t}^{\theta}}$ be its shifted counterpart. **Then, $\Delta x_t=\tilde{x_{t-1}} - x_{t}$ is negligible** where $\tilde{x_{t-1}} = \sqrt{\alpha_{t-1}}\ P_t(\tilde{\epsilon}_t^{\theta}(x_t)) + D_t(\tilde{\epsilon}_t^{\theta}(x_t))$.
> I.e., the shifted terms of $\tilde{\epsilon}_t^{\theta}$ in $P_t$ and $D_t$ destruct each other in the reverse process.
>
> >> Proof. **Define $\tilde{\epsilon_t^{\theta}} (x_t)=\epsilon_t^{\theta}(x_t)+\Delta\epsilon_{t}$, {$\beta_{t}$}$ㅤ_{t=1}^{T}$={$\beta_{1}$ $=\beta_{min}$, ..., $\beta_{T}=\beta_{max}$}, and $\alpha_{t}=\prod_{s=1}^{t} (1-\beta_s)$. Note that $\beta_{max}$ is defined as a small value (e.g., $\beta_{max} = 0.001$) and {$\beta_{t}$}$ㅤ_{t=1}^{T}$ are defined by a decreasing schedule from $\beta_T= \beta_{max}$ to $\beta_1 = \beta_{min} \approx 0$ (e.g., $\beta_{min} = 0.00001$).** Then, ...
> *****
> >[W2] The only quantitative evaluation is the user study. Results for DiffusionCLIP seem a bit lower quality than in the original paper.
>
> The attributes provided in the official repo produces the same quality.
> The lower quality results are from the facial expression changes where DiffusionCLIP struggles and does not provide checkpoints.
> The usage of checkpoint and training by ourselves are mentioned in the Appendix K.1. Below quotation provides reference for reducing burden to look up for them.
> >> We use official checkpoints provided by DiffusionCLIP except for some facial attributes whose checkpoint do not exist. We tried our best to tune their hyperparameters following the manual for fair comparison.
> The lower quality, which is not on the official repo and trained by us, shows that DiffusionCLIP is not good at facial expression change.
>
> Appendix K.2 provides quantitative comparison using segmentation consistency and directional CLIP similarity following DiffusionCLIP.
>
>
> >[W3] There are some minor language mistakes.
>
> The corrected mistakes in the updated version are marked in blue. Thanks.
>
> *****
>
> Let us mention a misunderstanding in the summary.
> > The loss function that they optimize is the same as for DiffusionCLIP.
>
> We use similar equation with directional CLIP loss and L1 loss but the arguments are different.
> - OURS
> 	- $directionalCLIP(P_t^{edit}, attr_{target},P_t^{original},attr_{source})$
> 	- $L1(x_t^{edit},x_t^{original})$
> - DiffusionCLIP - GPU efficient
> 	- $directionalCLIP(P_t^{edit}, attr_{target},x_0^{original},attr_{source})$
> 	- $L1(P_t^{edit},x_0^{original})$
> - DiffusionCLIP
> 	- $directionalCLIP(x_0^{edit}, attr_{target},x_0^{original},attr_{source})$
> 	- $L1(x_0^{edit},x_0^{original})$
>
> ******
> Thank you for the constructive comments. We hope these explanations resolve the concerns. Further questions or suggestions would be also appreciated.

---

> > ### Comment · Reviewer_KRCt · 2022-11-18
> > **Thanks**
> >
> > Thanks for the additional comments and clarifications. I do think it's a good paper and hope to see it published at ICLR 2023.

---

> > > ### Author Response · Authors · 2022-12-02
> > > **Thank you!**
> > >
> > > Thank you again for your suggestion and your great efforts. Please let us know any further questions/suggestions.

---

### Official Review · Reviewer_4Ber · 2022-10-24

**Confidence:** 4
**Correctness:** 3
**Technical Novelty And Significance:** 3
**Empirical Novelty And Significance:** 3
**Recommendation:** 6

**Clarity, Quality, Novelty And Reproducibility:**

- The paper is well-written for the most part.
- The contributions are significant and somewhat new. Aspects of the contributions exist in prior work.
- Results should be reproducible as the code is provided.

**Strength And Weaknesses:**

## strengths
- Finding a semantic latent space for pre-trained diffusion models is an important problem and is relatively less studied. The proposed Asyrp is a promising endeavor in this direction.
- The proposed h-space has several nice properties: homogeneity, linearity, robustness, and consistency across timesteps.
- The empirical study in generative process design suggests an interesting analysis for editing flexibility.

## weakness and detailed questions
- Theorem 1 seems a little bit weak to me, it says that $\Delta x_{t-1} \approx 0$ given a shift $\Delta \epsilon_t$. However, this conclusion is trivial given the well-known result from DDIM Eq. 13. The resulted loss function Eq. 7 seems same as the GPU-efficient finetuning in DiffusionCLIP [1].
- In Eq. 7, is $x_t^{edit}$ $P_t^{edit}$?
- Can we think of the proposed training of Asyrp as a constrained version of DiffusionCLIP (only the middle layer of UNet can be finetuned)? If so, how does the generative process of Asyrp compare to DiffusionCLIP, i.e. unmodified (symmetric) DDIM sampling?
- Linear combination (Figure 8) is quite interesting. How does this compare to a linear combination of scores (as in DiffusionCLIP)? This comparison would be interesting since it saves multiple UNet forward passes.
- This won't impact my rating. I am curious is it possible to extend the proposed h-space in text-to-image diffusion models such as StableDiffusion?

[1] Kim, Gwanghyun, Taesung Kwon, and Jong Chul Ye. 2021. “DiffusionCLIP: Text-Guided Diffusion Models for Robust Image Manipulation.” arXiv [cs.CV]. arXiv. http://arxiv.org/abs/2110.02711.


**Summary Of The Paper:**

This paper discovers a semantic latent space (termed h-space) for pretrained diffusion models. The proposed h-space is a shift/residual in the middle-layer feature of the UNet, and is discovered via finetuning. The authors propose asymmetric reverse process (Asyrp) for both finetune and sampling. The proposed h-space has several nice properties: homogeneity, linearity, robustness, and consistency across timesteps.

**Summary Of The Review:**

I am willing to amend my score is my major concerns are addressed.

---

> ### Author Response · Authors · 2022-11-09
> **Response to Reviewer 4Ber**
>
> Thank you for acknowledging the strengths of our paper: importance of the problem, nice properties of h-space, and analysis.
> We respectfully remind that _Asyrp does not finetune the U-Net_ in the diffusion models.
>
> *****
>
> >[W1] Theorem 1 seems a little bit weak to me, it says that $\Delta x_{t-1} \approx 0$ given $\Delta \epsilon_t$ a shift . However, this conclusion is trivial given the well-known result from DDIM Eq. 13.
>
> Theorem 1 implies that where the original $\epsilon_t$ leads to $x_{t-1}$  and  the edited $\tilde{\epsilon_{t}}=\Delta \epsilon_{t}+\epsilon_{t}$ leads to $\tilde{x_{t-1}}$, the  $\Delta x_{t-1}=x_{t-1} - \tilde{x_{t-1}} \approx 0$.  $\Delta x_{t-1}$ in our equation is different from $dx$ around DDIM Eq. 13.
>
> *****
> >[W2] The resulted loss function Eq. 7 seems same as the GPU-efficient finetuning in DiffusionCLIP [1].
>
> We use similar equation with directional CLIP loss and L1 loss but the arguments are different. Furthermore, the parameters being optimized are different: ours=$\Delta h$ or $f_t$, and DiffusionCLIP=U-Net.
> - OURS
> 	- $directionalCLIP(P_{t}^{edit},attr_{target},P_{t}^{original},attr_{source})$
> 	- $L1(x_t^{edit},x_t^{original})$
> - DiffusionCLIP - GPU efficient
> 	- $directionalCLIP(P_t^{edit}, attr_{target},x_0^{original},attr_{source})$
> 	- $L1(P_t^{edit},x_0^{original})$
> - DiffusionCLIP
> 	- $directionalCLIP(x_0^{edit}, attr_{target},x_0^{original},attr_{source})$
> 	- $L1(x_0^{edit},x_0^{original})$
> *****
> > [W3] In Eq. 7, is $x_t^{edit}$  $P_t^{edit}$?
>
> $P_t^{edit}$ is predicted $x_0$ from time step $t$, and $x_t^{edit}$ is the latent variable at the time step $t$.
> Please refer to Eq.3 and Eq.5.
>
> *****
> > [W4] Can we think of the proposed training of Asyrp as a constrained version of DiffusionCLIP (only the middle layer of UNet can be finetuned)? If so, how does the generative process of Asyrp compare to DiffusionCLIP, i.e. unmodified (symmetric) DDIM sampling?
>
> I am afraid we can't. Our method does _not_ finetune the U-Net but finds the semantic directions in h-space which are effective only with Asyrp in the frozen U-Net. Training the network representing the directions might be confused as finetuning U-Net but they are different. Please consider our method more like StyleCLIP which finds latent directions in a frozen StyleGAN.
>
> DiffuionCLIP requires one finetuned network for each attribute.
> In other words, it overfits the reverse process to one attribute. On the other hand, Asyrp enables h-space to be considered as a semantic latent space so that the same frozen network can edit different semantics through the found semantic directions, similarly to editing in $\mathcal{Z}$, $\mathcal{W}$, or $\mathcal{S}$ of GANs.
>
>
> *****
> > [Q1] Linear combination (Figure 8) is quite interesting. How does this compare to a linear combination of scores (as in DiffusionCLIP)? This comparison would be interesting since it saves multiple UNet forward passes.
>
> Instead of multiple reverse processes in DiffusionCLIP, we simply linearly add the semantic directions in h-space and perform a single same reverse process modified by the aggregated direction. It works just like how latent editing in GANs works.
>
>
> *****
> > [Q2] This won't impact my rating. I am curious is it possible to extend the proposed h-space in text-to-image diffusion models such as StableDiffusion?
>
> We observed similar phenomena in a few preliminary trials with StableDiffusion but we did not dive deep into it yet. We expect the other text-to-image diffusion models to show the same phenomena because they share the similar model in diffusion part. Unfortunately, their official codes are not released.
>
> Thank you for the constructive comments. We hope these explanations resolve the concerns. Further questions or suggestions would be also appreciated.

---

> ### Author Response · Authors · 2022-12-02
> **Response to Reviewer 4Ber**
>
> Again, thank you for your efforts and service. We would be appreciated it if you check out our revision of our paper according to your concerns. Please let us know if we have addressed all your concerns.
>
> Thank you.

---

> ### Author Response · Authors · 2022-12-10
> **Response to Reviewer 4Ber**
>
> Dear Reviewer 4Ber,
>
> Thank you again for the constructive feedback. This is a kind reminder of our response. We would greatly appreciate any subsequent thoughts on our efforts.
>
> Best regards,
> The authors

---

### Official Review · Reviewer_pRhF · 2022-10-26

**Confidence:** 4
**Correctness:** 2
**Technical Novelty And Significance:** 3
**Empirical Novelty And Significance:** 3
**Recommendation:** 6

**Clarity, Quality, Novelty And Reproducibility:**

Clarity: Could the authors discuss the relationship between CLIP-guided Diffusion and your theorem 1? It is a little bit weird to me since CLIP-guided diffusion does work, but your theory suggests the contrary?

Quality & Novelty: See the above section.

Reproducibility: The authors provide enough details in the appendix for reproduction. I think I can reproduce it by myself.

**Strength And Weaknesses:**

Strength:
(1) As far as I know, this paper is the first work to directly find the semantic directions in pre-trained diffusion models.
(2) The empirical results look impressive.

Weaknesses:
(1) The choice of h-space needs more clarification. The Figure 15 and Appendix C.3. could not explain everything. My questions are: [1] Are images in Figure 15 training images or test images? [2] If they are training images, why does the layers other than layer 8 have bad performance? I imagine they are all able to minimize the loss in Eq. 7. [3] If they are test images, why does layer 1-6 does not change the generated images at all? At least there should be some changes.
(2) The implicit neural direction looks unnecessary. Could the authors provide more ablation study to compare using $f_t$ and $h_t$? Moreover, I cannot understand the results in Figure 6 (a). Since $\epsilon$ has a higher dimension than $h$, I can hardly imagine it yields higher loss when optimizing Eq. (7).
(3) There are too many magic numbers in Section 4, making me doubt the generalizability of the proposed generative process. Though the authors claim that these magic numbers work across the five datasets, I still think quantitative results should be provided on analyzing the sensitivity of these magic numbers.


**Summary Of The Paper:**

The paper proposes asymmetric reverse process (Asyrp) to exploit the semantic latent space of pre-trained diffusion models. With the guidance of CLIP, the proposed method can find semantically meaningful directions in the latent space for image editing.

**Summary Of The Review:**

Overall, the goal of the paper is intriguing, but the paper contains a number of heuristics without enough strict ablation study and quantitative results, making the claims doubtful.

---

> ### Author Response · Authors · 2022-11-09
> **Response to Reviewer pRhF (2)**
>
> > [W3] There are too many magic numbers in Section 4, making me doubt the generalizability of the proposed generative process. Though the authors claim that these magic numbers work across the five datasets, I still think quantitative results should be provided on analyzing the sensitivity of these magic numbers.
>
> Our hyperparameters are far more principled than ones in DiffusionCLIP. They do not have any noticeable tendency ont the number of steps for the inversion and the coefficient of each losses across different for each datasets and each attributes. In contrast, we provide guidelines to choose hyperparameters with equations based on the tendency with measurable quantity such as CLIP similarity.
>
> We added Figure.22 and Figure.23 to visually analyze the effect of the hyperparameters.
>
> Thank you for the nice suggestion.
>
> *****
> > [Clarity] Could the authors discuss the relationship between CLIP-guided Diffusion and your theorem 1? It is a little bit weird to me since CLIP-guided diffusion does work, but your theory suggests the contrary?
>
> Let us settle the term first for safety. `Guidance` generally refers to the source signal that provides gradients to modify the trajectory of the reverse process by changing the latent variables themselves. DiffusionCLIP refers the CLIP loss as a `guidance` but it does not involve changing the latent variables but finetunes the U-Net according to the CLIP loss.
>
> Guidance-based approaches change the trajectory from $x_t$ to $x_{t-1}$ according to the gradient $\nabla$ from the guidance, e.g., CLIP or a pre-trained classifier, leading to different $\dot{x_{t-1}}=x_{t-1}+\nabla$. While $\nabla$ directly modifies the trajectory to $\dot{x_{t-1}}$, $\epsilon_t$ modifies $x_{t-1}$ through $P_t$ and $D_t$. Theorem 1 implies that the changes in $P_t$ and $D_t$ due to $\Delta \epsilon_t$ cancel out each other and thus $\tilde{x}_{t-1}$ barely changes.
>
> *****
> Thank you for the constructive comments. We hope these explanations resolve the concerns. Further questions or suggestions would be also appreciated.

---

> > ### Comment · Reviewer_pRhF · 2022-11-16
> > **Thank you!**
> >
> > Thanks for the clarification and revision of the paper. I've changed my score to a 6.

---

> > > ### Author Response · Authors · 2022-12-02
> > > **Thank you!**
> > >
> > > Again, thank you for your feedback and your great efforts. Any further questions/suggestions would be also appreciated.

---

> ### Author Response · Authors · 2022-11-09
> **Response to Reviewer pRhF (1)**
>
> Thank you for acknowledging our main contributions as the first attempt in the literature: asymmetric reverse process to make h-space as a semantic latent space of pre-trained diffusion models. We respectfully remind that, in contrast to guidance-based sampling methods, we use CLIP _only for finding semantic directions in h-space, not for reverse sampling_.
>
> *****
>
> > [W1] The choice of h-space needs more clarification. The Figure 15 and Appendix C.3. could not explain everything. My questions are:
> [W1-a] Are images in Figure 15 training images or test images?
> [W1-b] If they are training images, why does the layers other than layer 8 have bad performance? I imagine they are all able to minimize the loss in Eq. 7.
> [W1-c] If they are test images, why does layer 1-6 does not change the generated images at all? At least there should be some changes.
>
> [W1-a] Images in Figure 15 are training images. We chose the training images because to show that alternative choices lead to suboptimal results such as blur, noise, shape distortion, color distortion, or identity changes _even on the training images_. We added this in the caption.
>
> [W1-b, c] We suppose that the deepest layer of U-Net have the highest-level semantics as in any other architectures.
> Furthermore, we suppose that 1) the encoding part of U-Net extracts semantics from noisy inputs $\mathbf{x}_t$ in diffusion models and is robust to noise, and  2) the decoding part estimates $\epsilon_t$ by combining $\mathbf{h}$ and skip connections, especially the skip from the first layer where the lowest-level noise is captured. Hence, adding something to layer 1-6 barely changes the output and modifying intermediate features of layer 9-15 distorts $\epsilon_t$.
>
> We do not argue these ideas because they are just conjectures from observation and are difficult to be rigorously proved. Nevertheless, Figure 15 provides empirical grounds that the 8th layer is the only applicable semantic latent space.
>
> Please note that the results from other layers are the best among deliberate hyperparameter searching including learning rates, learning rate schedules, and weight decays.
>
> *****
> > [W2-1] The implicit neural direction looks unnecessary. Could the authors provide more ablation study to compare using ft and ht?
>
> Even though we can directly optimize $\Delta h_t$ or $\Delta h^{global}$, we opt to use $f_t$ because there are many advantages.
> We add the following in Appendix L.2 to provide the necessity of implicit neural direction.
> Thank you for the nice suggestion.
>
> >>Training time : $f_t \approx \Delta h^{global} < \Delta h_t$
> We have to optimize each $\Delta h_t$ for each time step $t$. Additionally, it needs specific hyperparameters for each $\Delta h_t$, e.g., higher learning rates for larger $t$. On the contrary, time-consumption for training $f_t$ is similar to optimizing $\Delta h^{global}$.
>
> >>Quality : $f_t \approx \Delta h_t > \Delta h^{global}$
> $f_t$ and $\Delta h_t$, where directions can be obtained for each timestep, have the best quality. As shown in Fig. 10, $\Delta h^{global}$ is sometimes accompanied by slight differences in hair, etc.
>
> >>Extensibility : $f_t > \Delta h_t > \Delta h^{global}$
> $\Delta h_t$ can be obtained from $f_t$, and $\Delta h^{global}$ can be obtained by aggregating $\Delta h_t$.
>
> >>We opt to use $f_t$ for above three advantages.
> *****
> >[W2-2] Moreover, I cannot understand the results in Figure 6 (a). Since $\epsilon$ has a higher dimension than $h$, I can hardly imagine it yields higher loss when optimizing Eq. (7).
>
> Even if $\epsilon$ has a higher dimension than $h$, $\epsilon$ has a dimension the same as the image, and it is the output of the network. We argue that it is rather difficult to make semantically meaningful changes because $\epsilon$ has a large dimension. For example, in the case of GANs, there are a few semantically meaningful changes when we modify high dimension features. Conversely, when we modify $z$ or $w$, semantic latent space, more semantically meaningful changes are shown.
>
> Experimentally, we assume that it is a phenomenon caused by $L1$ loss. In the case of optimizing $\Delta h_t$, we observed a gradual smile as the training progresses, while in the case of optimizing $\epsilon_t$, several artifacts are generated in the beginning, and artifacts gradually disappear as the training progresses. It is shown that less change occurs because of directly optimizing the values in pixel-level to satisfy $L1$ loss.
>
> ******

---

### Decision · Program_Chairs · 2023-01-20

**Decision:**

Accept: notable-top-25%

**Justification For Why Not Higher Score:**

As noted by the majority of reviewers, the theory is weak relative to the empirical results.

**Justification For Why Not Lower Score:**

All reviewers agreed that the empirical results are both extensive and quite strong. In addition to showing that the method can be used for different architectures across many datasets, the authors also conducted a user study to demonstrate performance superior to DiffusionCLIP.

**Metareview: Summary, Strengths And Weaknesses:**

The authors propose a method for finding a semantically meaningful latent space in pre-trained diffusion models. The idea is novel, the results are quite impressive, and work for many different architectures and datasets. One minor concern is that the theory (particularly Theorem 1) is a bit weak, but the utility of the method and the impressive empirical results outweigh weaknesses.

**Note From Pc:**

if the above contains the word "oral" or "spotlight" please see: "oral" presentation means -> notable-top-5% and "spotlight" means -> notable-top-25%. As stated in our emails, we are disassociating presentation type from AC recommendations